# Stability criteria for complex microbial communities

Stacey Butler[1] & James P. O'Dwyer[2,3]

Competition and mutualism are inevitable processes in microbial ecology, and a central question is which and how many taxa will persist in the face of these interactions. Ecological theory has demonstrated that when direct, pairwise interactions among a group of species are too numerous, or too strong, then the coexistence of these species will be unstable to any slight perturbation. Here, we refine and to some extent overturn that understanding, by considering explicitly the resources that microbes consume and produce. In contrast to more complex organisms, microbial cells consume primarily abiotic resources, and mutualistic interactions are often mediated through the mechanism of crossfeeding. We show that if microbes consume, but do not produce resources, then any positive equilibrium will always be stable to small perturbations. We go on to show that in the presence of crossfeeding, stability is no longer guaranteed. However, positive equilibria remain stable whenever mutualistic interactions are either sufficiently weak, or when all pairs of taxa reciprocate each other's assistance.

[1] Department of Mathematics, University of Illinois, Urbana, IL 61801, USA. [2] Department of Plant Biology, University of Illinois, Urbana, IL 61801, USA. [3] Carl R. Woese Institute for Genomic Biology, University of Illinois, Urbana, IL 61801, USA. Correspondence and requests for materials should be addressed to J.P.O. (email: jodwyer@illinois.edu)

arly ecological intuition suggested that tightly woven networks of interactions would lead to more stable and robust communities[1–3]. But this intuition was later overthrown by the realization that large, complex, interacting systems will tend to become unstable to small perturbations once either the number or strength of interactions passes a certain threshold[4,5]. These original analyses were based on the strength and type of interactions between pairs of species, and initially pertained to random mixtures of interaction types, including predation, competition, and mutualism. More recently, similar results were derived for purely competitive interactions among a group of species, with the same bottom line: if pairwise competitive interactions are too numerous, or too strong, then any equilibrium of coexisting, competing species will be unstable[6].

In parallel, theory had also been developed for groups of species consuming a set of distinct resources[3,7–12]. In these models, the resources actually being competed for are treated explicitly. Typically these models make several simplifying assumptions, including the timescales on which resources are replenished, and the way that consumer preferences differ. Some treat resources as biological organisms, with the capacity to grow and compete among themselves[7,13], while others consider abiotic resources, which are replenished from outside the system[14]. Microbial communities composed of bacterial or archael cells provide an example of an ecological system where we can (almost) unambiguously separate the component parts into biological organisms and abiotic resources, where by abiotic we mean resources that are not capable of reproduction on their own. Even though a rich range of metabolites exchanged by microbial communities are of biological origin, we count these as abiotic for this reason. Direct predatory interactions among bacteria are somewhat rare[15], while consumption and production of abiotic resources likely mediates much of microbial competition (via resource scarcity[16]) and mutalism (via crossfeeding[17]), suggesting that a consumer–producer–resource framework will provide a more general and appropriate framework for microbial interactions than direct, pairwise interactions. However, the broad-ranging properties of such large systems of consumers, producers, and abiotic resources are under-explored.

With these motivations in mind, we present a model of consumers that compete for a set of dynamical, abiotic resources, governed by a set of preferences for those resources, an influx rate for resource replenishment, and a mortality rate for consumers. We will prove that any positive densities of consumers and resources can be an equilibrium solution in our model, with no finite limit to the similarity of consumers. Moreover, in contrast to results for pairwise competition, these equilibria are always stable to small perturbations, so long as there are at least as many resources as consumers. We place no specific constraint on the set of preferences of consumers for resources to obtain this result—in other words, we do not need to assume a particular functional form (neither random, modular, or highly structured in some other way). For a fixed set of preferences of consumers for resources, we also use this model to examine the range of influx and mortality rates that lead to stable coexistence. We find that pairwise species similarity alone is not sufficient to determine the size of this range, thus clarifying and refining the classical expectation of limiting similarity in this context[3,9,18,19].

Finally, we extend our model to include production of resources. Mutualistic pairwise interactions have been found to push communities closer to instability[6], leading to a debate over how widespread mutualistic interactions can reasonably be in microbial communities[20,21]. We introduce a model for the exchange of resources, where microbes can both consume and now also produce resources, and choose a form for these equations that is interpretable as either a species-specific leakage of

resources, or as a kind of recycling of biomass following mortality. In this model, we find a similar result in our consumer–producer–resource system, and for a range of cases bound the possible strength and prevalence of resource production. On the other hand, we find that if mutalistic interactions are completely symmetric, then stability is again guaranteed for feasible equilibria, a result at odds with earlier pairwise analyses.

## Results

**Model of competition for abiotic resources.** Our model for consumers and resources is defined in terms of competition for substitutable resources:

$$
\begin{cases}
\dot{R}_i = \rho_i - R_i \sum_j C_{ij} S_j, \\
\dot{S}_i = \epsilon S_i \sum_j C_{ij}^T R_j - \mu_i S_i.
\end{cases}
\tag{1}
$$

Here, $i$ can take any value from 1 to $N$, the $R_i$ represent a set of $N$ resources, while the $S_i$ are a set of $N$ consumers, and the non-negative quantity $C_{ij}$ is the rate of consumption of resource $i$ by consumer $j$, per unit consumer and resource. $\rho_i$ and $\mu_j$ represent influx rates for resource $i$ and mortality rate for consumer $j$, respectively, while $\epsilon$ is a free parameter characterizing the efficiency with which consumers convert resources into biomass. We could include an outflow, or leaching rate $-\eta_i R_i$ for resources[14], though here we assume that consumption is sufficiently fast that this rate will be negligible (and moreover our results below for local stability are unchanged even when this outflow is incorporated). We also note that we could straightforwardly generalize these equations to include more general functional responses[22] (for example it would be perfectly reasonable to consider a Monod-type form for the uptake rate of resources). However, we consider solely the mass action terms above in the spirit of the vast range of Lotka–Volterra analyses undertaken for pairwise interactions: if we can understand the properties of these idealized communities, we then have a baseline on which to layer further biological complexity.

Finally, we note that this is a model for substitutable resources, and while there may be families of resources which are to some extent substitutable (for example different carbon sources) the general picture for microbial consumers is likely colimitation by multiple, qualitatively different types of resource[14,23,24]. In many cases, we might expect that only one of these resources is actually rate-limiting (roughly, the rarest in a given environmental context), and this assumption leads to Liebig's law of the minimum[25], where growth rate of a consumer depends only on this single resource. In other circumstances, two or more resources may turn out to be limiting in any given environmental context, in which case growth rate has typically has been modeled as proportional to the product of these limiting resources, and termed multiplicative colimitation[24]. Beyond these phenomenological approximations of colimitation, there might be still more general functional dependencies[26]. Taking these possibilities into account, our analysis of substitutable resource consumption and production here may be seen as a starting point for these more general cases, and a good approximation for circumstances where a single 'type' or family of resources is rate-limiting.

The consumer–resource system represented by (1) has a set of equilibrium solutions

$$
\begin{cases}
\vec{R}^* = (C^T)^{-1} \frac{1}{\epsilon} (\vec{\mu}), \\
\vec{S}^* = \left[ (\vec{R}^*_{\text{diag}}) C \right]^{-1} \vec{\rho}.
\end{cases}
\tag{2}
$$

where $\vec{R}^*_{\text{diag}}$ denotes a diagonal matrix with the components of $\vec{R}^*$

along the diagonal. The only biologically reasonable solutions are those with positive densities of consumers and resources, known as feasible solutions[27]. For our model, for any positive $\{\vec{R}^*, \vec{S}^*\}$, there are always positive influx and mortality rates $\vec{\rho}$ and $\vec{\mu}$ that result in these feasible solutions, independent of the form of the matrix $C$ (our Methods Section details the proof of this statement). In other words, any feasible set of densities can be a solution of our equations, for an appropriate set of influx and mortality rates defining the environmental context. We now present two results. First, we will show that feasible solutions are stable to small perturbations. Then we will consider what range of parameter values for $\rho_i$ and $\mu_i$ will lead to such feasible solutions.

**Feasibility and stability under competitive interactions**. We now demonstrate a departure from earlier results for pairwise competiton between species[6]: any and all of these feasible solutions are guaranteed to be stable to small perturbations. The Jacobian matrix corresponding to the equilibrium solutions given by (2) is

$$L = \left[ \begin{array}{c|c} [-C\vec{S}^*]_{diag} & -[\vec{R}^*_{diag}]C \\ \hline \epsilon[\vec{S}^*_{diag}]C^T & 0. \end{array} \right]. \quad (3)$$

If all the eigenvalues of this Jacobian have negative real part, then this equilibrium is locally stable. In our Methods Section, we show that for a feasible solution (i.e., an equilibrium with positive abundances) this equilibrium is guaranteed to be stable to local perturbations, independent of the form of the matrix $C$, implying that competition alone in systems of consumers and resources is never sufficient to lead to an unstable, feasible equilibrium. We illustrate this (along with the full distribution of eigenvalues) for some example cases in Fig. 1. These plots demonstrate visually the form of the spectrum across a range of cases (both generalist and specialist consumers, and ordered and unordered resources). Beyond the fact that the largest eigenvalue always has negative real part, as our results state, we also note that the spectra have a characteristic 'dragonfly' form. The wings of the dragonfly will lead to the potential for oscillatory behavior around this equilibrium, while the typically large density of real eigenvalues near zero imply that some types of perturbation away from these equilibria will have long relaxation times.

We now note several important generalizations. First, including the leaching or degradation of resources, which would ensure that resources would saturate even in the absence of consumers, does not change the form of this matrix or the result that feasibility implies stability. Second, we have so far considered equal numbers of resources and consumers. We can generalize this result to incorporate uinequal numbers of resources and consumers, $N_R$ and $N_S$, and in our Methods Section we show that feasibility only implies stability in this case if $N_R \geq N_S$, mirroring the classical expectation that coexistence of $N_S$ species will require at least that number of resources. Finally, we generalize this result to the case where one or more consumer has zero abundance. In our Methods Section we show that the remaining group of consumers (those with positive abundances) will coexist stably so long as the equilibrium is uninvasible by any of the extirpated consumers.

These results bear comparison with earlier calculations for the stability of systems of biotic consumers and biotic resources[13,28]. In these systems, resources themselves can grow and compete with each other, and can be eaten by the consumers. When treated as a set of Lotka–Volterra equations, in those cases feasibility also implies stability. The similarity with our results suggests that the horizontal structure here (i.e., a clear division of the system into consumers and resources) is key element for the

local stability guarantee to hold. In large systems of consumers and biotic resources it might be seen as artificial to completely separate consumers and resources into two groups. On the other hand, in microbial systems with a clear biological distinction between abiotic resources and biotic consumers, this separation is very natural, and so our result applies unambiguously and generally.

**Structural stability under competitive interactions**. We now ask what range of values of $\vec{\mu}$ and $\vec{\rho}$ will lead to positive, feasible solutions for consumer and resource densities, for a given fixed set of preferences for resources, $C$. The volume of this parameter space is known as the structural stability of a given system[29], and biologically it quantifies the robustness of equilibria. Suppose that the environmental context in which a group of species coexist shifts over time, and this shift affects $\vec{\mu}$ and $\vec{\rho}$. Then structural stability characterizes how likely it will be that the same species will continue to coexist in this new environmental context. In our Methods Section we derive results for the volume of $\vec{\mu}$ and $\vec{\rho}$ values that will lead to feasible solutions, for a given matrix $C$. For example, in the case of mortality rates, this volume is given by

$$V_\mu = 2\frac{|\det C|}{\pi^{\frac{n}{2}}} \int_{\mathbb{R}^n_+} e^{-\langle \xi, C^T C \xi \rangle} d\xi, \quad (4)$$

where $\det C$ is the determinant of the matrix $C$ and the angled brackets define the usual dot product. This contrasts with what perhaps might be the intuitive determinant of structural stability —something like the average of pairwise species similarity (say, defined in terms of the overlaps of two consumers' preferences). In contrast, the determinant here does not depend in a simple way on the similarity of any two species—structural stability depends on $C$ as a whole, rather than being a function just of pairwise similarities.

We can still ask how this volume changes as we make consumers more or less similar in terms of the resources they use, represented mathematically by the inner product of the columns of $C$. For example, in the special case where all species begin equally similar, then a uniform decrease in their similarity leads to a larger volume for the parameter space, and greater structural stability (detailed in Methods). This is in agreement with the ecological intuition that it is 'easier' for more dissimilar species to coexist. However, this is not the general case. Independent of the size of the system, there are some contexts where a decrease in the similarity of any pair of species will lead to less, not greater, structural stability. Figure 2 shows a specific example, for three species, where a decrease in the similarity of one pair of species (while keeping other pairwise similarities fixed) leads to a decrease in structural similarity. What is the general lesson here? The biological interpretation is that changes in structural stability as we change the similarity of any two species depend (in general) on all of the other consumption preferences. This reiterates our point above, that the determinant in $V_\mu$ does not depend in a trivial way on pairwise comparisons, and consequently the structural stability of systems of consumers and abiotic resources does not depend in a simple way on pairwise species similaritities expressed in terms of consumption preferences.

**Model of consumption and exchange of resources**. Microbial taxa that can both consume and also produce resources have the potential for mutualistic, syntrophic interactions[6,20,30]. For this reason we now consider a more general set of dynamics for species and resources, which includes a matrix, $C$, representing the resource use of the consumers and a nonnegative matrix, $P$,

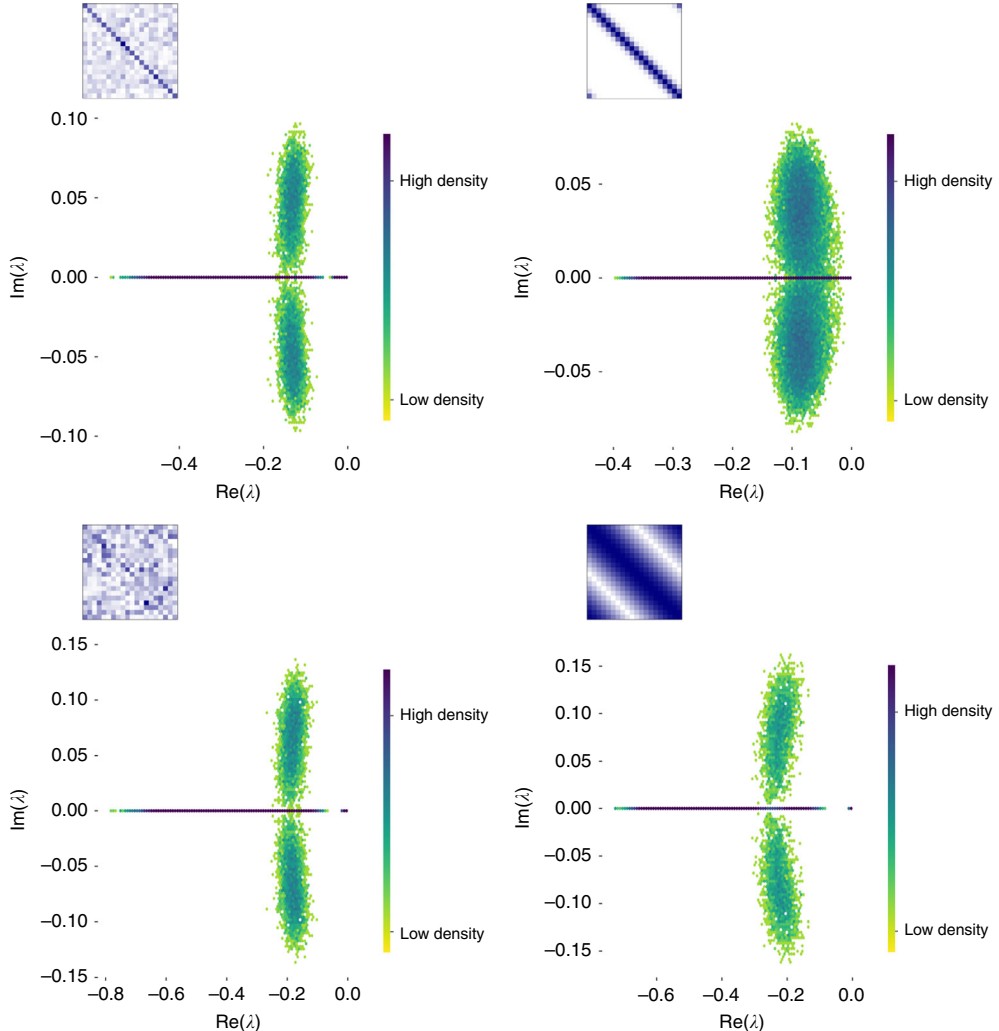

**Fig. 1** Positive equilibria are guaranteed to be stable under competition for abiotic resources. We show four examples demonstrating feasible solutions of (1) that are stable to small perturbations. Plots show the density (colored from red to blue) of eigenvalues for the Jacobian matrix at this equilibrium, defined using a fixed matrix of consumer preferences. The form of the consumer preferences is shown inset in blue, where each row represents a distinct resource, each column represents a distinct consumer, and darker blue indicates a higher rate of consumption. Each plot is obtained over multiple random draw for consumer and resource densities, drawn from a uniform distribution. In the left-hand panels, we consider a gradient from near-specialism, where each consumer has a favorite resource (but there are weak, randomly-drawn off-diagonal interactions), to near-generalism, where the off-diagonal preferences are stronger than the specialism. In the right-hand panels, we show a similar gradient of near-specialism to near-generalism, but where the resource preferences follow a smooth curve away from the preferred resource for each species. Both left- and right-hand panels therefore show a gradient from generalism to specialism, but the right-hand case assumes that there is an unambigious spectrum of similarity for resources, and that species that can consume a given resource also tend to consume similar resources. In all four cases, our theorem for local stability holds: the real parts of all eigenvalues of the Jacobian matrix are always negative. We also note the similarity in the 'dragonfly' shape for this distribution across all cases, contrasting with the classic circle (or elliptical) distributions for eigenvalues found in the case of pairwise interactions[6], but similar to the distribution of eigenvalues found for bipartite Lotka–Volterra equilibria[39]

representing the production of resources by the consumers:

$$
\begin{cases}
\dot{R}_i = \rho_i - R_i \sum_j C_{ij} S_j + \sum_j P_{ij} S_j, \\
\dot{S}_i = \epsilon S_i \sum_j C_{ij}^T R_j - S_i \sum_j P_{ij}^T - \mu_i S_i.
\end{cases}
\tag{5}
$$

Our approach again makes use of mass action principles, and we note that we consider production of resources to remove biomass from the consumer density, meaning that terms corresponding to this leakage appear in both equations. We have also chosen a form for production that depends solely on the density of species that are present in the system, and not on what resources those species are using to grow. This may be a reasonable approximation for some processes, for example the

production of various intermediates of the TCA cycle, the production of which are not substrate dependent[31]. But in other cases, we may need to allow for a more general form for production that depends both on the species that are present, and the specific resources they are consuming. Another interpretation of the same terms would be as a kind of recycling—where mortality does not just remove consumers from the system, but also returns some portion of their biomass to the common resource pool.

We now consider the properties of the equilibrium solutions

$$
\begin{cases}
\vec{R}^* = (C^T)^{-1} \frac{1}{\epsilon} (P^T \vec{1} + \vec{\mu}), \\
\vec{S}^* = \left[ (R_{\text{diag}}^*) C - P \right]^{-1} \vec{\rho}.
\end{cases}
\tag{6}
$$

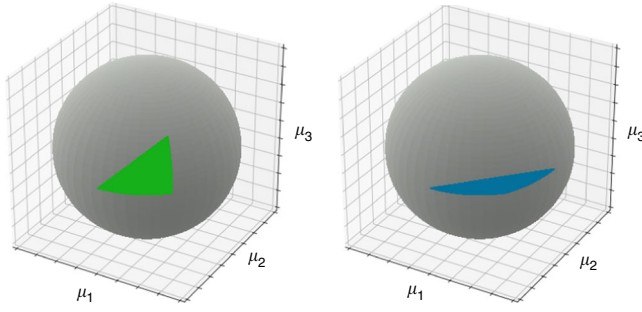

**Fig. 2** Structural stability changes non-monotonically with species similarity. The volume of the set of mortality rates leading to feasible densities for the resources, $R^*$, can decrease even when species similarity is decreased. Here, we show an example in three dimensions, where each axis represents one of the three mortality rates, $\mu_i$, and the volume is a kind of wedge extending from the origin outwards. The measure, $V_\mu$, of the size of this volume is then equivalent to the area (colored green or blue) of a triangle on the surface of the unit sphere, where the dissimilarity of each pair of species is proportional to the length of one of the triangle's sides. On the left, this volume is shown for the particular $3 \times 3$ matrix $C^T$ detailed in our Methods section, and is colored green. When the angle between one pair of column vectors is increased while the other angles are unchanged, we get the volume shown on the right-hand side. The resulting volume decreases in size, despite the average similarity of these three species having decreased

which have a correspondingly more general Jacobian matrix

$$L = \left[ \begin{array}{c|c} [-CS^*]_{\text{diag}} & -[R^*_{\text{diag}}]C + P \\ \hline \epsilon[S^*_{\text{diag}}]C^T & 0 \end{array} \right]. \quad (7)$$

For this system, when the production matrix is nonzero, we do not know in general what additional constraints beyond feasibility are needed to ensure a stable equilibrium. However, we present evidence below that suggests production rates must in general be bounded for feasible solutions to also be locally stable. Conversely, we also show that there are some cases in which production rates can be arbitrarily large and local stability will still hold.

**Feasibility and reciprocity guarantee local stability**. To explore this, we consider the case of specialist consumers, $C = cI$, i.e., where each consumer specializes on just a single resource. We also tune the influx and mortality rates so that equilbrium species abundances and resource densities take the simplified form $\vec{S}^* = s\vec{1}$, and $\vec{R}^* = r\vec{1}$, for positive real values $s$ and $r$. While this may seem restrictive, in that competition is solely intraspecific, we put no other restrictions on the production rates, $P_{ij}$. Hence, we can think of this as close to the most general case of purely mutualistic interspecific interactions. In our Methods Section we express constraints on the production matrix to ensure that it is possible to obtain these feasible solutions. We then go on to prove that a sufficient (though not necessary) condition for feasible solutions to also be stable is

$$\left( \sum_{j \neq i} P_{ij} \right)^2 < \frac{cs}{\epsilon} \left( cr - P_{ii} - \frac{cs}{4\epsilon} \right). \quad (8)$$

This result is straightforward to interpret. Suppose that consumers do not produce any of the resource that they specialize on, so that $P_{ii}$ vanishes. Then this result constrains a combination of the strength and prevalence of production of the

'off-diagonal' resources for each species in terms of consumption rate, $c$, conversion efficiency, $\epsilon$, and the equilibrium abundances and densities $s$ and $r$ of consumers and resources. These results are reminiscent of constraints on stability for randomly drawn, pairwise interactions between species. In that case, mutualistic interactions tend to be more destabilizing than competitive interactions. Are we just recapitulating results that are qualitatively already understood via pairwise interactions? We emphasize that here our bound is sufficient, but not necessary for feasibility to imply stability, and there are cases where the bound above is not necessary for stability. One example is when production vanishes altogether, and we are back to our earlier, more general result that feasibility always implies stability.

A second example is when the production matrix is purely symmetric. In this case, feasibility alone is sufficient to guarantee local stability of the equilibrium solution, with no further restrictions on production. We call this case reciprocal, because symmetry of $P$ ensures that each (specialized) consumer gives exactly as much of each other consumer's preferred resource as they receive from that consumer. Figure 3 demonstrates this for some particular examples. The importance of reciprocity for stability seemingly contrasts with analysis of mutualistic interactions using direct interactions between species—for example in the case of plant-pollinator networks[32]. In these highly structured communities, it was found that strong effects of plants on pollinators (or vice versa) must come with weak interactions of pollinators on plants, or else the resulting community equilibrium would tend to be unstable. Even though these communities differ from the random interactions considered by ref. [6] and others, the same principle lies behind these results: if overall interaction strengths exceed a bound determined by intraspecific regulation, then a positive equilibrium can be unstable. In ref. [32], this bound means that stability requires a balance of strong and weak interactions, leading to asymmetry, rather than reciprocity. Our finding that the asymmetry in mutualistic interactions can be destabilizing rather than stabilizing demonstrates the quite different insights to be gained from the analysis of consumer–producer–resource communities.

**Discussion**
We have modeled the interactions among biological populations that can consume, produce, and reproduce; and abiotic resources that can be introduced, leached, consumed, or produced, but do not reproduce. Perhaps more than any other biological system, this clear division is likely to be a good approximation in bacterial and archaeal communities, and may provide a more accurate description than modeling pairwise competitive and mutualistic interactions, which ignore the dynamics of resources. We revisit a series of classic analyses for ecological communities in this framework, and identify important differences with earlier theoretical results arising from direct, pairwise interactions. First, we find that any positive densities for consumers and resources can be an equilibrium solution to our equations, given an appropriate environmental context. We also find that these feasible solutions are always locally stable, unlike the classical results for pairwise interactions between species, which allow for unstable, feasible solutions[5,6] unless particular restrictions are placed on species interactions[27,33].

For systems of consumers and resources, we derived results for structural stability given a set of consumer preferences, a measure of robustness to environmental changes. In contrast to the classical expectation, we showed that structural stability is not guaranteed to increase as species become more dissimilar in terms of their resource preferences, echoing other recent work showing the complexity of structural stability for direct, pairwise

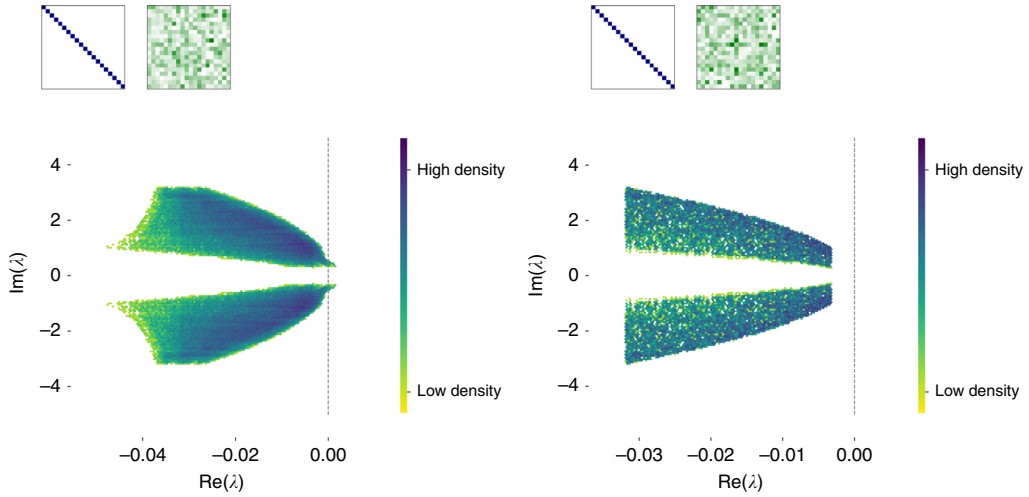

**Fig. 3** Positive equilibria are guaranteed to be stable if exchange of resources is reciprocal. We show two examples demonstrating feasible solutions of (5). Plots show the density (colored from red to blue) of eigenvalues for the Jacobian matrix at this equilibrium, defined using a fixed, diagonal matrix of consumer preferences. This is shown inset in blue, where each row represents a distinct resource and each column represents a distinct consumer; blue squares indicates a nonzero rate of consumption. We also consider a fixed, more general matrix of production rates (inset in green). Again, each row represents a distinct resource, each column represents a distinct consumer, and darker green indicates a higher rate of production. Each plot is obtained over multiple random draws for consumer and resource densities s and r, defined in the main text, and drawn from a uniform distribution (subject to the constraints necessary to ensure that these densities can be obtained with positive influx and mortality rates $\rho$ and $\mu$). In the left-hand panel, we consider a random set of production rates, which does not satisfy the bound necessary to guarantee stability, and indeed we see that there are some positive eigenvalues of the Jacobian matrix, to the right of the black dashed line. In the right-hand panel, we show a similar case but where we impose that the production matrix P is symmetric, meaning that each consumer gets as much of its favored resource as it gives. Even though the production matrix looks 'similar' to the naked eye in each case, this symmetry in the latter example is enough to guarantee local stability, with the largest eigenvalue bounded away from zero by a gap related to the consumer abundances

interactions[27,33]. Our results clarify how this more general picture plays out when resource dynamics are modeled explicitly. Finally, we extended our approach to include the production of resources, which allows for mutualistic interactions via crossfeeding, where one species may produce a resource that another needs. In this case, we find that when production is too large, feasible solutions no longer guarantee local stability. But we also find that when mutualistic interactions are precisely balanced, that stability again is guaranteed. If species reciprocate with each other cooperatively, positive equilibria will be locally stable.

There are several opportunities for the extension of our results: including the consideration of more complex functional responses[22,34] allowing for the saturation of resource usage or multi-way interactions; demographic or environmental stochasticity with (for example) their effects on local extirpation of resources or consumers[35]; and extending the more general analyses of global stability that have been developed in the case of Lotka–Volterra competition and mutualism[13,36] to this system of consumers and abiotic resources. We also do not rule out that direct, pairwise interactions are likely to play a role (perhaps as an approximation to antagonistic interactions mediated by antiobiotic production, or even in rare cases direct cannibalism of microbial cells by each other), and clearly more general forms of colimitation and production are both possible and likely relevant in real communities. Finally, in the cases we have analyzed, we have been able to prove that there are circumstances when feasibility implies stability. But even in these cases, there is more to investigate in terms of the form of the eigenvalue spectrum. Exact or approximate solutions for the full spectrum, including what controls its overall shape and the size of its largest eigenvalues, will shed light on the dynamics near equilibrium.

In summary, our results reflect a general lesson. In any complex ecological system we inevitably draw lines around what we choose to model, and what we leave out. Our results here show

that explicitly modeling the resources that mediate interactions between biological organisms can significantly alter our conclusions regarding stability, and hence what kinds of community structure we can expect to observe in nature.

## Methods

**Feasibility of solutions.** For any solution for $\vec{R}^*$ and $\vec{S}^*$ with positive consumer and resource densities, we can identify positive-valued influx and mortality rates that will lead to these solutions as follows:

$$\begin{cases} \vec{\mu} = \epsilon_{\text{diag}} C^T \vec{R}^*, \\ \vec{\rho} = \vec{R}^*_{\text{diag}} C \vec{S}^*. \end{cases} \quad (9)$$

Because C is positive, any positive $\vec{R}^*$ will result in $\vec{\mu} > 0$. And if $\vec{S}^*$ is positive, $\vec{\rho} > 0$. Thus for appropriate choice of parameters, any positive $\{\vec{R}^*, \vec{S}^*\}$ can be found as a solution to Eq. (1) in the main text.

**Stability of feasible equilibria for competition.** The Jacobian matrix corresponding to the equilibrium solutions given by Eq. (2) in the main text is

$$L = \begin{bmatrix} [-C\vec{S}^*]_{\text{diag}} & -[\vec{R}^*_{\text{diag}}]C \\ \hline \epsilon[\vec{S}^*_{\text{diag}}]C^T & 0 \end{bmatrix}. \quad (10)$$

If all the real parts of the eigenvalues of this Jacobian are negative, then this equilibrium is locally stable. To compute these eigenvalues, we first note that the eigenvalue equation $\det(L - \lambda I) = 0$ is given by

$$\det(L - \lambda I) = \det\left(\begin{bmatrix} [-C\vec{S}^*]_{\text{diag}} - \lambda I & -[\vec{R}^*_{\text{diag}}]C \\ \hline \epsilon[\vec{S}^*_{\text{diag}}]C^T & -\lambda I \end{bmatrix}\right) = 0. \quad (11)$$

Next, we note that $\left[[-C\vec{S}^*]_{\text{diag}} - \lambda I\right]$ is invertible, so long as $\lambda$ is not an eigenvalue of $[-C\vec{S}^*]_{\text{diag}}$. If it is an eigenvalue, then for nonnegative, invertible C and positive $\vec{S}^*$, the entries of $C\vec{S}^*$ are positive. And so for $\lambda$ an eigenvalue of $[-C\vec{S}^*]_{\text{diag}}$, we have $\lambda < 0$. On the other hand, if $\lambda$ is not an eigenvalue of

$[-C\vec{S}^*]_{\text{diag}}$ then

$$
\begin{aligned}
\det(L - \lambda I) &= \det\left([-C\vec{S}^*]_{\text{diag}} - \lambda I\right) \\
&\times \det\left(-\lambda I - \epsilon\left[\vec{S}^*_{\text{diag}}\right]C^T\left[[-C\vec{S}^*]_{\text{diag}} - \lambda I\right]^{-1}\left[-\left[\vec{R}^*_{\text{diag}}\right]C\right]\right).
\end{aligned}
$$

Defining $D_1 = \epsilon\left[\vec{S}^*_{\text{diag}}\right]$, $D_2 = \left[[-C\vec{S}^*]_{\text{diag}} - \lambda I\right]^{-1}\left[-\left[\vec{R}^*_{\text{diag}}\right]\right]$ and $k = \det\left([-C\vec{S}^*]_{\text{diag}} - \lambda I\right)$ then

$$
\begin{aligned}
\det(L - \lambda I) &= k\det(-\lambda I - D_1 C^T D_2 C) \\
&= k\det\left(-\lambda I - [\sqrt{D_2}C\sqrt{D_1}]^T[\sqrt{D_2}C\sqrt{D_1}]\right). \tag{12}
\end{aligned}
$$

We now assume $\lambda$ is a nonnegative real number. Then so long as $D_1$, $D_2$, and $C$ are invertible (true by previous assumption and with $\vec{S}^*$, $\vec{R}^* > 0$, then $[\sqrt{D_2}C\sqrt{D_1}]^T[\sqrt{D_2}C\sqrt{D_1}]$ is positive definite and thus $\lambda < 0$, contradicting the assumption. Biologically, our assumption that $C$ is invertible only rules out essentially trivial cases, where e.g. a resource (or a linear combination of resources) can be removed from the system altogether, or the resource preferences of two species overlap identically.

More generally, assume $\text{Re}(\lambda) \geq 0$. From (12) we have

$$
\begin{aligned}
0 &= \det(-\lambda I - D_1 C^T D_2 C) \\
&= \det(-\lambda I - \sqrt{D_1}C^T D_2 C\sqrt{D_1}). \tag{13}
\end{aligned}
$$

We wish to find a contradiction by proving the eigenvalues of $\sqrt{D_1}C^T D_2 C\sqrt{D_1}$ have positive real part and so it will be sufficient to show that the Hermitian part, $H(\sqrt{D_1}C^T D_2 C\sqrt{D_1})$ is positive definite.

$$
\begin{aligned}
H(\sqrt{D_1}C^T D_2 C\sqrt{D_1}) &= \tfrac{1}{2}\left(\sqrt{D_1}C^T D_2 C\sqrt{D_1} + \sqrt{D_1}C^T D_2^\dagger C\sqrt{D_1}\right) \\
&= \sqrt{D_1}C^T \text{Re}(D_2)C\sqrt{D_1} \\
&= \left[\sqrt{\text{Re}(D_2)}C\sqrt{D_1}\right]^T\left[\sqrt{\text{Re}(D_2)}C\sqrt{D_1}\right] \tag{14}
\end{aligned}
$$

Note that $\text{Re}(D_2) > 0$, and $\sqrt{\text{Re}(D_2)}$, $C$ and $\sqrt{D_1}$ are all invertible and so $H(\sqrt{D_1}C^T D_2 C\sqrt{D_1})$ is positive definite. Thus, we must have that $\text{Re}(\lambda) < 0$.

Equivalently, we can give a more explicit proof, where we assume that $\lambda = c + di$ is complex with real part $c \geq 0$. Then we can choose the $j$th diagonal entry of $\sqrt{D_2}$ to be written as $a_j + b_j i$ where $|a_j| > |b_j|$ for all $j$, because each of the entries of the diagonal matrix $D_2$ has positive real part. Now, let $\sqrt{D_2}C\sqrt{D_1} = A + Bi$ with $A = \vec{a}_{\text{diag}}C\sqrt{D_1}$ and $B = \vec{b}_{\text{diag}}C\sqrt{D_1}$. Let $\vec{x}$ be an eigenvector associated with the eigenvalue $-\lambda$ of $[A + Bi]^T[A + Bi]$. Then $\vec{x}^*[A + Bi]^T[A + Bi]\vec{x}$ should have non-positive real part; however,

$$
\begin{aligned}
\vec{x}^*[A + Bi]^T[A + Bi]\vec{x} &= \vec{x}^* A^T A\vec{x} - \vec{x}^* B^T B\vec{x} \\
&+ [\vec{x}^* A^T B\vec{x} + \vec{x}^* B^T A\vec{x}]i \tag{15}
\end{aligned}
$$

and the real part of $\vec{x}^*[A + Bi]^T[A + Bi]\vec{x}$ is

$$
\vec{x}^* A^T A\vec{x} - \vec{x}^* B^T B\vec{x} \tag{16}
$$

and because

$$
A\vec{x} = \vec{a}_{\text{diag}}C\sqrt{D_1}\vec{x} = \vec{a}_{\text{diag}}\vec{r} \tag{17}
$$

and

$$
B\vec{x} = \vec{b}_{\text{diag}}C\sqrt{D_1}\vec{x} = \vec{b}_{\text{diag}}\vec{r} \tag{18}
$$

where $\vec{r} = C\sqrt{D_1}\vec{x}$, we have that $\vec{x}^* A^T A\vec{x} - \vec{x}^* B^T B\vec{x} \geq 0$. If $\vec{x}^* A^T A\vec{x} - \vec{x}^* B^T B\vec{x} = 0$, then it must be true that $\vec{r} = 0$. Because $C$ is assumed to be invertible, this means that $\sqrt{D_1}\vec{x} = 0$; however, this is not possible based on the definitions of $D_1$ and $\vec{x}$. Therefore we contradict our initial assumption that $\lambda$ has nonnegative real part.

**Structural stability for competition.** Feasible solutions for this system correspond to positive values for $\vec{R}^*$ and $\vec{S}^*$. The size of $\mu$-space that lead to such positive solutions then depends on $C$, while similarly the size of $\rho$-space depends on $C^T$. The size of the parameter space depends on $\det(C)$ which considers the relationship among all consumers' resource use vectors and so it is more than just pairwise similarity that is important for the size of the parameter spaces[33,37]. This can be

seen from mathematical expressions for these volumes, which are given by[37]

$$
V_\mu = 2\frac{\sqrt{\det M_\mu}}{\pi^{\frac{n}{2}}}\int_{\mathbb{R}^n_+} e^{-\langle\xi, M_\mu\xi\rangle}\,\mathrm{d}\xi \tag{19}
$$

and

$$
V_\rho = 2\frac{\sqrt{\det M_\rho}}{\pi^{\frac{n}{2}}}\int_{\mathbb{R}^n_+} e^{-\langle\xi, M_\rho\xi\rangle}\,\mathrm{d}\xi. \tag{20}
$$

Here, $M_\mu = \hat{C}\hat{C}^T$ and the columns of $\hat{C}^T$ are the normalized columns of $C^T$ and $M_\rho = \tilde{C}^T\tilde{C}$ and the columns of $\tilde{C}$ are the normalized columns of $R^*C$. We note that while $V_\mu$ depends only on the properties of the consumer preference matrix, $V_\rho$ depends also on $\vec{R}^*$, which in turn depends on a specific choice of $\vec{\mu}$. In other words, the volume $V_\rho$ changes with choice of $\vec{\mu}$.

In the special case where all angles between pairs of column vectors are initially the same, a uniform increase in all angles does lead to a larger volume for the parameter space. To see this for a system with $N$ consumers, let $a = \cos(\theta)$ where $\theta$ represents the angle between each pair of normalized column vectors of $C$. The determinant of $M$ is given by

$$
\begin{aligned}
\det(M) &= \det\begin{pmatrix} 1 & a & \cdots & a \\ a & 1 & \ddots & \vdots \\ \vdots & \ddots & \ddots & a \\ a & \cdots & a & 1 \end{pmatrix} \\
&= (1 - a)^{n-1}(1 + (n - 1)a). \tag{21}
\end{aligned}
$$

Thus if the angle, $\theta$, increases, then $a = \cos(\theta)$ decreases and $\det(M)$ will increase.

However, it is not always the case that increasing angles between one or more species will increase $V_\mu$ or $V_\rho$. We now consider the following example for $N = 3$, shown graphically in Fig. 2. Suppose initially that consumer preferences are given by the matrix

$$
C^T = \begin{bmatrix} 1 & \frac{1}{\sqrt{2}} & \frac{1}{\sqrt{3}} \\ 0 & \frac{1}{\sqrt{2}} & \frac{1}{\sqrt{3}} \\ 0 & 0 & \frac{1}{\sqrt{3}} \end{bmatrix}, \tag{22}
$$

while after shifting species preferences we have the new matrix

$$
\tilde{C}^T = \begin{bmatrix} 1 & \frac{1}{\sqrt{2}} & a \\ 0 & \frac{1}{\sqrt{2}} & b \\ 0 & 0 & c \end{bmatrix}, \tag{23}
$$

where $a = \frac{1}{\sqrt{3}} - \frac{1}{4}\sqrt{\frac{194}{75}}$, $b = \frac{1}{\sqrt{3}} + \frac{1}{4}\sqrt{\frac{194}{75}}$, and $c = \frac{1}{10}$. The angles between the column pairs for $C^T$ are $\{\theta_{12} = \frac{\pi}{4}, \theta_{13} \approx 0.304\pi, \theta_{23} \approx 0.196\pi\}$ and the angles between the column pairs for $\tilde{C}^T$ are $\{\tilde{\theta}_{12} = \frac{\pi}{4}, \tilde{\theta}_{13} \approx 0.444\pi, \tilde{\theta}_{23} \approx 0.196\pi\}$. So that $\theta_{12} = \tilde{\theta}_{12}$, $\theta_{13} < \tilde{\theta}_{13}$ and $\theta_{23} = \tilde{\theta}_{23}$. The volumes of the associated parameter spaces are $V_\mu \approx 0.0417$ and $\tilde{V}_\mu \approx 0.008$ for $C^T$ and $\tilde{C}^T$, respectively. We note that these are just exemplars, and this counterintuitive result can be found for many different contexts, and is not limited to three species.

**Feasibility and stability for competition and crossfeeding.** For the case of specialized consumers and equilibria with equal abundances for species and equal densities for all resources, the equations for equilibrium abundances are (from Eq. (6) in the main text)

$$
\begin{cases} r\vec{1} = \frac{1}{c\epsilon}\left(P^T\vec{1} + \vec{\mu}\right), \\ s\vec{1} = [crI - P]^{-1}\vec{\rho}. \end{cases} \tag{24}
$$

Then, solving for the parameters $\vec{\mu}$ and $\vec{\rho}$ we have

$$
\begin{cases} \vec{\mu} = cr\epsilon\vec{1} - P^T\vec{1}, \\ \vec{\rho} = s[crI - P]\vec{1}. \end{cases} \tag{25}
$$

To ensure that positive resource density (i.e., positive $r$) can be obtained by positive rates, $\vec{\mu}$, we need that $cr\epsilon > \sum_j P^T_{ij}$, for all $i$. Similarly, to ensure that we obtain positive values for $s$ from positive $\rho$, we need that $cr > \sum_j P_{ij}$ for all $i$. In summary, not all possible values of consumer and resource density (i.e., not all positive values of $r$ and $s$) can be obtained as solutions of these equations, conditioning on positive influx and mortality rates. That is, unlike the case of $P = 0$, there are now constraints for what subsets of feasible solutions are possible.

From Eq. (7) in the main text, we then find the Jacobian reduces to the form

$$\tilde{L} = \left[ \begin{array}{c|c} -csI & P - crI \\ \hline \epsilon csI & 0 \end{array} \right]. \tag{26}$$

We let $\lambda$ represent an eigenvalue of $\tilde{L}$. Then, $\gamma = \frac{\lambda(\lambda+cs)}{\epsilon cs} + cr$ is an eigenvalue of $P$. Hence,

$$\lambda = \frac{-cs}{2} \pm \frac{1}{2}\sqrt{(cs)^2 - 4\epsilon cs(cr - \gamma)}.$$

When all $\text{Re}(\lambda) < 0$ the equilibria above are stable to local perturbations.

We first consider a special case: where all $\gamma$ are all real. An example of this occurs when $P$ is symmetric, when (as discussed in the main text) each consumer species will get as much of its preferred resource as it gives to other species (of their preferred resources). In this case, if also all $\gamma < cr$, then all $\lambda$ will be $< 0$ and the equilibrium is locally stable. Using Gershgorin's theorem[38], this will be guaranteed when

$$cr > \sum_j P_{ij} \quad \forall i$$

In fact, this is the same as the criterion above that ensures $\vec{S}^* > 0$. So we conclude that feasibility and reciprocity together imply local stability, visualized in an example case of Fig. 3 in the main text.

To find necessary and sufficient conditions on $\gamma$ ensuring this for non-symmetric $P$, first let $\hat{\gamma} = |\hat{\gamma}|e^{i\theta} = cs - 4\epsilon(cr - \gamma)$. Then we want

$$\begin{array}{rcl} \text{Re}\left(\sqrt{\hat{\gamma}}\right) & < & \sqrt{cs}, \\ \sqrt{|\hat{\gamma}|}\cos\frac{\theta}{2} & < & \sqrt{cs}, \\ |\hat{\gamma}|\left(\frac{1+\cos\theta}{2}\right) & < & cs, \\ |\hat{\gamma}| + \text{Re}(\hat{\gamma}) & < & 2cs \end{array} \tag{27}$$

or equivalently

$$\epsilon(Im(\gamma))^2 < cs(cr - \text{Re}(\gamma)). \tag{28}$$

Can we reexpress this in terms of direct criteria for $P$ that will guarantee stability? Again using Gershgorin's theorem[38], for each eigenvalue, $\gamma$, of $P$ we know that for some $i$

$$|\gamma - P_{ii}| < \sum_{j\neq i} P_{ij} \tag{29}$$

or equivalently:

$$(Im(\gamma))^2 < -(\text{Re}(\gamma))^2 + 2P_{ii}\text{Re}(\gamma) + \left(\sum_{j\neq i} P_{ij}\right)^2 - (P_{ii})^2 \tag{30}$$

We can now rewrite our criteria for local stability as

$$\epsilon\left(-(\text{Re}(\gamma))^2 + 2P_{ii}\text{Re}(\gamma) + \left(\sum_{j\neq i} P_{ij}\right)^2 - (P_{ii})^2\right) < cs(cr - \text{Re}(\gamma)). \tag{31}$$

Rearranging terms and maximizing the quadratic in $\text{Re}(\gamma)$ gives

$$\left(\sum_{j\neq i} P_{ij}\right)^2 < \frac{cs}{\epsilon}\left(cr - P_{ii} - \frac{cs}{4\epsilon}\right) \tag{32}$$

which is sufficient for stability when it holds for all $i$.

**Unequal numbers of consumers and resources**. For any solution for $\vec{R}^*$ and $\vec{S}^*$ with positive consumer and resource densities where the number of consumers is $N_S$ and the number of resources is $N_R$, with $N_S$ not necessarily equal to $N_R$, we can identify positive-valued influx and mortality rates that will lead to these solutions as follows:

$$\begin{cases} \vec{\mu} = \epsilon_{\text{diag}} C^T \vec{R}^*, \\ \vec{\rho} = \vec{R}^*_{\text{diag}} C \vec{S}^*. \end{cases} \tag{33}$$

Because $C$ is positive, any positive $\vec{R}^*$ will result in $\vec{\mu} > 0$. And if $\vec{S}^*$ is positive, $\vec{\rho} > 0$. Thus for appropriate choice of parameters, any positive $\{\vec{R}^*, \vec{S}^*\}$ can be found as a solution to Eq. (1) in the main text.

Let the number of consumers be $N_S$, and the number of resources be $N_R$. Assume an equilibrium solution such that all resources and all consumer species have positive density. The Jacobian matrix corresponding to any such equilibrium

solution is

$$L = \left[ \begin{array}{c|c} [-C\vec{S}^*]_{\text{diag}} & -[\vec{R}^*_{\text{diag}}]C \\ \hline \epsilon[\vec{S}^*_{\text{diag}}]C^T & 0 \end{array} \right]. \tag{34}$$

If all the real parts of the eigenvalues of this Jacobian are negative, then this equilibrium is locally stable. To compute these eigenvalues, we first note that the eigenvalue equation $\det(L - \lambda I) = 0$ is given by

$$\det(L - \lambda I) = \det\left(\left[ \begin{array}{c|c} [-C\vec{S}^*]_{\text{diag}} - \lambda I & -[\vec{R}^*_{\text{diag}}]C \\ \hline \epsilon[\vec{S}^*_{\text{diag}}]C^T & -\lambda I \end{array} \right]\right) = 0. \tag{35}$$

Next, we note that $\left[[-C\vec{S}^*]_{\text{diag}} - \lambda I\right]$ is invertible, so long as $\lambda$ is not an eigenvalue of $\left[-C\vec{S}^*\right]_{\text{diag}}$. If it is an eigenvalue, then the entries of $C\vec{S}^*$ are nonnegative and so $\lambda \leq 0$. However, if for some $j$, $(C\vec{S}^*)_j = 0$, then from Eq. (1), there is no nontrivial, finite equilibrium value for the corresponding resource, $R_j$. Thus, we consider this a degenerate situation and assume the entries of $C\vec{S}^*$ to be strictly positive. And so for $\lambda$ an eigenvalue of $\left[-C\vec{S}^*\right]_{\text{diag}}$, we have $\lambda < 0$. On the other hand, if $\lambda$ is not an eigenvalue of $\left[-C\vec{S}^*\right]_{\text{diag}}$ then

$$\begin{aligned} 0 &= \det(L - \lambda I) \\ &= \det\left(\left[-C\vec{S}^*\right]_{\text{diag}} - \lambda I\right) \\ &\quad \times \det\left(-\lambda I - \epsilon\left[\vec{S}^*_{\text{diag}}\right]C^T\left[\left[-C\vec{S}^*\right]_{\text{diag}} - \lambda I\right]^{-1}\left[-\left[\vec{R}^*_{\text{diag}}\right]C\right]\right). \end{aligned}$$

Note that $\det\left(\left[-C\vec{S}^*\right]_{\text{diag}} - \lambda I\right) \neq 0$. Thus, the above equation implies

$$0 = \det\left(-\lambda I - \epsilon\left[(\vec{S}^*)_{\text{diag}}\right]C^T\left[\left[-C\vec{S}^*\right]_{\text{diag}} - \lambda I\right]^{-1}\left[-\left[\vec{R}^*_{\text{diag}}\right]C\right]\right). \tag{36}$$

Defining $D_1 = \epsilon\left[(\vec{S}^*)_{\text{diag}}\right]$ and $D_2 = \left[\left[-C\vec{S}^*\right]_{\text{diag}} - \lambda I\right]^{-1}\left[-\left[\vec{R}^*_{\text{diag}}\right]\right]$ then

$$\begin{aligned} 0 &= \det(-\lambda I - D_1 C^T D_2 C) \\ &= \det(-\lambda I - \sqrt{D_1}C^T D_2 C\sqrt{D_1}). \end{aligned} \tag{37}$$

Assume $\text{Re}(\lambda) \geq 0$. We wish to find a contradiction by proving the eigenvalues of $\sqrt{D_1}C^T D_2 C\sqrt{D_1}$ have positive real part and so it will be sufficient to show that the Hermitian part, $H(\sqrt{D_1}C^T D_2 C\sqrt{D_1})$ is positive definite.

$$\begin{aligned} H(\sqrt{D_1}C^T D_2 C\sqrt{D_1}) &= \tfrac{1}{2}\left(\sqrt{D_1}C^T D_2 C\sqrt{D_1} + \sqrt{D_1}C^T D_2^\dagger C\sqrt{D_1}\right) \\ &= \sqrt{D_1}C^T \text{Re}(D_2)C\sqrt{D_1} \\ &= \left[\sqrt{\text{Re}(D_2)}C\sqrt{D_1}\right]^T\left[\sqrt{\text{Re}(D_2)}C\sqrt{D_1}\right]. \end{aligned} \tag{38}$$

Note that $\text{Re}(D_2) > 0$. Now $\sqrt{\text{Re}(D_2)}C\sqrt{D_1}$ is an $N_R \times N_S$ dimensional matrix. We claim that $\text{rank}\left(\sqrt{\text{Re}(D_2)}C\sqrt{D_1}\right) = N_S$. First, because the entries of $\sqrt{\text{Re}(D_2)}$ and $\sqrt{D_1}$ are positive, $\text{rank}\left(\sqrt{\text{Re}(D_2)}C\sqrt{D_1}\right) = \text{rank}(C)$. Now we assume $\text{rank}(C) = N_S$. This is possible only when $N_S \leq N_R$, in keeping with the expectation that coexistence of a group of $N_S$ species requires at least that many resources. Viewed another way, this means that the consumer species have sufficiently unique resource preferences. Now, with $\text{rank}\left(\sqrt{\text{Re}(D_2)}C\sqrt{D_1}\right) = N_S$, we may conclude that $H(\sqrt{D_1}C^T D_2 C\sqrt{D_1})$ is positive definite.

Next, let the number of consumers be $N_S$, and the number of resources be $N_R$. Assume an equilibrium solution such that all resources have positive density, and all consumer species have nonnegative density. The Jacobian matrix corresponding to any such equilibrium solution is

$$L = \left[ \begin{array}{c|c} [-C\vec{S}^*]_{\text{diag}} & -[\vec{R}^*_{\text{diag}}]C \\ \hline \epsilon[\vec{S}^*_{\text{diag}}]C^T & [\epsilon C^T R^* - \mu]_{\text{diag}} \end{array} \right]. \tag{39}$$

If all the real parts of the eigenvalues of this Jacobian are negative, then this equilibrium is locally stable. To compute these eigenvalues, we first note that the eigenvalue equation $\det(L - \lambda I) = 0$ is given by

$$\det(L - \lambda I) = \det\left(\left[ \begin{array}{c|c} [-C\vec{S}^*]_{\text{diag}} - \lambda I & -[\vec{R}^*_{\text{diag}}]C \\ \hline \epsilon[\vec{S}^*_{\text{diag}}]C^T & [\epsilon C^T R^* - \mu]_{\text{diag}} - \lambda I \end{array} \right]\right) = 0. \tag{40}$$

Next, we note that $\left[[-C\vec{S}^*]_{\text{diag}} - \lambda I\right]$ is invertible, so long as $\lambda$ is not an eigenvalue of $\left[-C\vec{S}^*\right]_{\text{diag}}$. If it is an eigenvalue, then the entries of $C\vec{S}^*$ are

nonnegative and so $\lambda \leq 0$. However, if for some $j$, $\left(C\vec{S}^*\right)_j = 0$, then from Eq. (1), there is no nontrivial, finite equilibrium value for the corresponding resource, $R_j$. Thus, we consider this a degenerate situation and assume the entries of $C\vec{S}^*$ to be strictly positive. And so for $\lambda$ an eigenvalue of $\left[-C\vec{S}^*\right]_{\text{diag}}$, we have $\lambda < 0$. On the other hand, if $\lambda$ is not an eigenvalue of $\left[-C\vec{S}^*\right]_{\text{diag}}$ then

$$0 = \det(L - \lambda I)$$
$$= \det\left(\left[-C\vec{S}^*\right]_{\text{diag}} - \lambda I\right)$$
$$\times \det\left(\left[\epsilon C^T R^* - \mu\right]_{\text{diag}} - \lambda I - \epsilon\left[\vec{S}^*_{\text{diag}}\right]C^T\left[\left[-C\vec{S}^*\right]_{\text{diag}} - \lambda I\right]^{-1}\left[-\left[\vec{R}^*_{\text{diag}}\right]C\right]\right).$$

Note that $\det\left(\left[-C\vec{S}^*\right]_{\text{diag}} - \lambda I\right) \neq 0$. For simplicity, assume that there is only one consumer with non-positive density, $S^*_j = 0$. For all other consumers, the corresponding entries of $\left[\epsilon C^T R^* - \mu\right]_{\text{diag}}$ are zero. Thus, the above equation implies

$$0 = \left(\left[\epsilon C^T R^* - \mu\right]_j - \lambda\right)$$
$$\times \quad \det\left(-\lambda I - \epsilon\left[(\vec{S}^*_m)_{\text{diag}}\right]C_m^T\left[\left[-C_m\vec{S}^*_m\right]_{\text{diag}} - \lambda I\right]^{-1}\left[-\left[\vec{R}^*_{\text{diag}}\right]C_m\right]\right) \quad (41)$$

With

$$\vec{S}^*_m = \vec{S}^* \text{ with the } j\text{th entry removed}$$
$$C_m = C \text{ with the } j\text{th column removed}$$

Defining $D_1 = \epsilon\left[(\vec{S}^*_m)_{\text{diag}}\right]$ and $D_2 = \left[\left[-C_m\vec{S}^*_m\right]_{\text{diag}} - \lambda I\right]^{-1}\left[-\left[\vec{R}^*_{\text{diag}}\right]\right]$ then

$$0 = \left(\left[\epsilon C^T R^* - \mu\right]_j - \lambda\right) \times \det\left(-\lambda I - D_1 C_m^T D_2 C_m\right)$$
$$= \left(\left[\epsilon C^T R^* - \mu\right]_j - \lambda\right) \times \det\left(-\lambda I - \sqrt{D_1}C_m^T D_2 C_m\sqrt{D_1}\right) \quad (42)$$

So we have that $\left[\epsilon C^T R^* - \mu\right]_j$ is an eigenvalue. Now looking at $0 = \det\left(-\lambda I - \sqrt{D_1}C_m^T D_2 C_m\sqrt{D_1}\right)$, assume $\text{Re}(\lambda) > 0$. We wish to find a contradiction by proving the eigenvalues of $\sqrt{D_1}C_m^T D_2 C_m\sqrt{D_1}$ have positive real part and so it will be sufficient to show that the Hermitian part, $H\left(\sqrt{D_1}C_m^T D_2 C_m\sqrt{D_1}\right)$ is positive definite.

$$H\left(\sqrt{D_1}C_m^T D_2 C_m\sqrt{D_1}\right) = \frac{1}{2}\left(\sqrt{D_1}C_m^T D_2 C_m\sqrt{D_1} + \sqrt{D_1}C_m^T D_2^\dagger C_m\sqrt{D_1}\right)$$
$$= \sqrt{D_1}C_m^T \text{Re}(D_2) C_m\sqrt{D_1}$$
$$= \left[\sqrt{\text{Re}(D_2)}C_m\sqrt{D_1}\right]^T\left[\sqrt{\text{Re}(D_2)}C_m\sqrt{D_1}\right] \quad (43)$$

Note that $\text{Re}(D_2) > 0$. Now $\sqrt{\text{Re}(D_2)}C_m\sqrt{D_1}$ is an $N_R \times N_S - 1$ dimensional matrix. We claim that $\text{rank}\left(\sqrt{\text{Re}(D_2)}C_m\sqrt{D_1}\right) = N_S - 1$. First, because the entries of $\sqrt{\text{Re}(D_2)}$ and $\sqrt{D_1}$ are positive, $\text{rank}\left(\sqrt{\text{Re}(D_2)}C_m\sqrt{D_1}\right) = \text{rank}(C_m)$. Now we will have $\text{rank}(C_m) = N_S - 1$ if $\text{rank}(C) = N_S$. This is only possible when $N_S \leq N_R$. This is again a biologically reasonable assumption to make, as it means that the consumer species have sufficiently unique resource preferences. Now, with $\text{rank}\left(\sqrt{\text{Re}(D_2)}C_m\sqrt{D_1}\right) = N_S - 1$, we may conclude that $H\left(\sqrt{D_1}C_m^T D_2 C_m\sqrt{D_1}\right)$ is positive definite. Thus the stability of the equilibrium $\{\vec{R}^*, \vec{S}^*\}$ depends only on the sign of $\left[\epsilon C^T R^* - \mu\right]_j$. If $\left[\epsilon C^T R^* - \mu\right]_j < 0$, the system is uninvasible by species $j$ and the equilibrium is stable.

**Code availability**. The authors used python code to generate matrices of appropriate structure to produce Figs. 1–3. The code necessary to generate these figures (or related) will be made available in the O'Dwyer lab github repository (https://github.com/odwyer-lab/consumer_producer).

**Data availability**. No datasets were generated or analyzed during the current study.

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

## Acknowledgements

J.O.D. acknowledges the Simons Foundation Grant #376199, McDonnell Foundation Grant #220020439, NSF #DEB1557192

## Author contributions

S.B. and J.O.D. contributed equally to this work.

## Additional information

**Competing interests:** The authors declare no competing interests.

