## [Peer Review File · Nature Communications]

Reviewers' comments:

Reviewer #1 (Remarks to the Author):

This is a well-crafted, interesting manuscript analyzing a model composed of populations competing for abiotic resources. The analysis is well done and complete, the text is brief and easy to follow. The supplement contains important information and is well-organized.

I have only a few comments/questions.

1. The model studied in the manuscript is one where production is decoupled from consumption. The Authors comment on this fact when describing Eq. 5. This is a limitation of the study, but I believe that the Authors are being upfront.
2. When production is large, the effect is destabilizing, unless production is symmetric. The Authors suggest that then the asymmetry, rather than the strength, is destabilizing. This is interesting in light of Bascompte et al. (Science 2006), who argued precisely the opposite. This part could be expanded, and the analysis contrasted with that of Bascompte et al. (who also used LV-like dynamics).
3. The distribution of eigenvalues in Figure 1 is quite interesting. My impression is that the real eigenvalues spreading on the x-axis must be strongly correlated with those of the diagonal matrix $-CS^*$. The "wings" of the dragonfly seem to be similar to those found for purely bipartite systems by Grilli et al. (Nature Comm, 2016), shifted to the left. In particular, the method of Grilli et al. could be used to approximate the span of the "wings".
4. The form taken by the Jacobian is very similar to the case studied by Case and Casten (American Naturalist, 1979). I believe the Authors should comment on the similarities/differences with this study. Also the study of Goh (also American Naturalist, 1979) should be discussed.

Overall, I greatly enjoyed reading the manuscript.

Stefano Allesina

Reviewer #2 (Remarks to the Author):

The subject of this paper is the stability of equilibrium solutions to systems of resource-consumer equations modelling microbial communities. Many classical results treat ecological as pairs of species that are directly interacting, either via competitive, mutualistic, predator-prey interactions, or a combination thereof. In reality, competitive and mutualistic interactions are frequently not direct but mediated via, respectively, consumption of a shared resource, or production by one species of a resource that is consumed by another. This paper takes the classical approach of random matrix models for ecological communities and investigates the effect of treating interactions not as direct pairwise interactions, but explicitly as consumer-resource interactions. The authors show that this changes some of the results from the classical theory. Specifically, they show that there is no loss of stability with increasing complexity when interactions are only via consumption. When there are also interactions via production, this can cause a loss of stability and this depends on factors such as the asymmetry of the effective mutualistic interactions induced by crossfeeding.

Overall, I found the paper to be clearly written, the results to be novel and sound and the interpretation appropriate. The authors strike a reasonable balance of making the paper accessible to

a non-specialist yet including enough detail in the Supplementary Material. I recommend publication and have only minor comments:

1. The authors focus on the local stability of feasible equilibria, which they define as solutions where all species have strictly positive density. There are also a raft of solution in which one or more species has zero density. These can be accommodated within the theoretical framework as they can be viewed as a strictly positive solution of a subsystem of the original model, with the zero species removed. However, this masks an important question which is how does species coexistence depend on model parameters ("complexity"). If an increase in complexity tends to cause species extirpations (via a transcritical bifurcation in which a species' equilibrium density becomes zero), this will not be detected by the current analysis. This is a common feature of this type of theoretical framework, but the authors should at least add a discussion of this limitation of their approach.
2. The text on p2 should be labelled as an Introduction section.
3. In eq. 2 the notation C^{-T} is ambiguous. I assume it means the inverse of the transpose of C. If so this can be unambiguously and better denoted as $(C^T)^{-1}$
4. The definition at top of p4 of what is meant by the "diag" notation should be moved to immediately after eq. 2 where the notation first appears.
5. P4: "If all eigenvalues of this Jacobian are negative" → "If all eigenvalues of this Jacobian have negative real part"
6. p4 "Independent of the form of the matrix C". I am guessing it is assumed that all elements of C are non-negative. This assumption should be explicitly stated either here or at eq. 1
7. I found the 2nd paragraph of p4 confusing. How are the earlier results less valid than the current ones? Some more detail in the argument is needed here.
8. In the text discussing Fig. 1 it would be helpful to provide a description of the biological interpretation of the 4 different types of structure tested for the matrix C.
9. Following eq. 5 state that element of P are non-negative
10. The sufficient (but not necessary) condition in eq. 8 comes from an upper bound on the real part of the eigenvalues. Upper bounds can be misleading! For example, taken at face value eq. 8 might be taken as suggesting that increases in P or decreases in the RHS of eq. 8 will tend to destabilise the system. But it perfectly possible for the upper bound on the eigenvalue to be increasing as P increases, but for the eigenvalue itself to be decreasing. On the whole, the authors use appropriately cautious language in interpreting their result in eq. 8. But it would be worth adding an explicit caveat that it is quite possible for systems to become more stable (i.e. the dominant eigenvalue's real part to decrease) not less as production rates increase.

Reviewer #3 (Remarks to the Author):

Understanding the conditions in which microbial communities will be stable is an important problem in community ecology. In this paper, the authors present analytical results of a resource-consumer model and its variant. In its basic form, the model is defined in Eqs. (1) and accounts for different consumer preference (C_{ij}), different resource influx rates (ρ_j) and consumer death rate (μ_i). In the basic model, the species do not produce resources. In its variant (Eq. 5) mutualistic interactions are considered.

The authors show that in the basic model 1): Feasible solutions are stable regardless of the dimension of the system N (feasible solutions mean that there is a fixed point with positive values for the number of resources and consumers). 2) The volume (= "number", aka structural stability) of feasible solutions may (= for particular parameter instances) decrease even when species similarity is decreased (Fig. 2). Finally, the authors incorporate cross-feeding and derive sufficient conditions on the matrix of cross-feeding interactions for which the feasible solutions will be stable (Eq 8).

The significance of these results is the fact that they differ from those obtained in pairwise-interaction models (see e.g. Allesina et al. [6]) where the resource-consuming interactions are not accounted for. I am not an expert on this class of theoretical models, so I am not able to judge the novelty of some parts of the work (although the first part of the paper feels like something that should have been done before... it would be good for the authors to be more clear about this). I thought that the model was a generally reasonable model and was analyzed in a rigorous way, although I often felt that intuition was not provided for the reader. In addition, the figures were not very helpful in supporting the authors' paper. Given the issues specified above I sometimes had difficulty fully appreciating the paper, but at the same time I believe that this sort of resource-explicit model has the potential to provide tremendous insight into microbial communities.

MAJOR POINTS

- It is my understanding that the authors' results are valid only in the case $N \text{ resources} = N \text{ consumers}$. Is that true? If so, I'd prefer to have this point expressed more clearly in the introduction, and perhaps show in the Discussion section whether or not the results generalize to the case $\text{no. resources} \neq \text{no. consumers}$. This is particularly important given the history of the field in which these issues have received considerable attention.
- I feel that the 'Results' section does not add much, beyond stating the results. For example, it is not very helpful to show Eq. (3) or (4) because those formulae do not become clear until one gets to the 'Method' section. In contrast, the Method section is highly technical and, in its current stage, not suitable for the average reader. I'd recommend the authors to expand the 'Results' section and provide some intuition about their results.
- The authors' model assumes that resources are made in proportion to the species abundance rather than their growth (or what they are growing on). This is a major simplifying assumption, so the authors should perhaps be more clear about it.
- I do not think that Fig. 2 is very helpful in supporting the authors' argument. Maybe it'd be better to compute (numerically) and the volume V_{μ} as a function of the number of species (for the parameter instance for which the effect is observed)? It is not immediately clear from the figure that the blue volume is smaller than the red one.
- In general, it seems that the figure purpose is to exemplify the claims by instantiating the model for certain parameter values. This is OK, but not sufficient for me to believe that the claims are correct. Reproducing the calculations might not be possible for every reader given that they are highly-technical. I think the authors should concentrate their efforts in either finding a biological explanation, or to corroborate their results further with numerical simulations.

MINOR POINTS

- Sec. 1.1 'characterizing' typos
- I would recommend to add $i = 1, \dots, N$ next to Eq. (1).
- Sec. 1.1 par 2: The authors may want to explain 'Liebig's law of minimum' and the other rules that define limiting factors of microbial growth. I think this sentence contains too much jargon.
- Fig. 1 and Fig. 3 It'd be helpful to label the axes of the insets. For example, in Fig. 1, it is my understanding that the insets represent $N \times N$ matrices whose i, j - th element defines preference of i -th

consumer for j -th resource (or the other way around)? Blue indicates greater preference than white (or the other way around)? Are they just the C matrices?

- The main result conveyed in Fig. 1 is the fact that $\text{Re}[\lambda] < 0$. I was wondering if it'd help to plot $\max \text{Re}[\lambda]$ against a control parameter that spans from generalism to specialism.

- Fig. 1 What are the implications in the dynamics in having a 'dragonfly' shape of the spectrum compared to the classical ellipses?

- Fig. 1 and 3: could the authors add a legend for the green-red colors used?

Response to Reviews for **Stability Criteria for Complex Microbial Communities**

Stacey Butler¹,
James O'Dwyer^{2,3*},

1 Department of Mathematics, University of Illinois, Urbana, IL 61801, USA

2 Department of Plant Biology, University of Illinois, Urbana, IL 61801, USA

3 Carl R. Woese Institute for Genomic Biology, University of Illinois, Urbana, IL 61801, USA

RESPONSE TO EDITORIAL COMMENTS

There were no specific editorial comments to respond to, other than focusing on a point-by-point response to reviewers, which we present below.

RESPONSE TO INDIVIDUAL REVIEWS

Response to Referee 1's Comments for the Authors

This is a well-crafted, interesting manuscript analyzing a model composed of populations competing for abiotic resources. The analysis is well done and complete, the text is brief and easy to follow. The supplement contains important information and is well-organized.

We thank the reviewer for the complimentary comments and are glad that the central ideas came across clearly.

I have only a few comments/questions.

1. *The model studied in the manuscript is one where production is decoupled from consumption. The Authors comment on this fact when describing Eq. 5. This is a limitation of the study, but I believe that the Authors are being upfront.*

This is a fair comment. We think our interpretation of the production terms is biologically-reasonable—either as a species-specific form of leakage, or as a form of recycling where some portion of consumer biomass is returned to the common pool during mortality. On the other hand, we completely acknowledge that the form of these terms could be open to further generalization.

To address this comment, we modified text in two places. Below the introduction of the equations involving leakage (in the subsection titled ‘Mutualistic Interactions’) we added to and modified the following paragraph:

“Our approach again makes use of mass action principles, and we note that we consider production of resources to remove biomass from the consumer density, meaning that terms corresponding to this leakage appear in both equations. We have also chosen a form for production that depends solely on the density of species that are present in the system, and not on what resources those species are using to grow. This may be a reasonable approximation for some processes, for example the production of various intermediates of the TCA cycle, the production of which are not substrate-dependent. But in other cases, we may need to allow for a more general form for production that depends both on the species that are present, *and* the specific resources they are consuming. Another interpretation of the same terms would be as a kind of recycling—where mortality doesn’t just remove consumers from the system, but also returns some portion of their biomass to the common resource pool.”

We also thought it was worthwhile outlining these choices in the introduction, and so we added this sentence to the fourth paragraph of the introduction:

“We introduce a model for the exchange of resources, where microbes can both consume and now also produce resources, and choose a form for these equations that is interpretable as either a species-specific leakage of resources, or as a kind of recycling of biomass following mortality.”

2. *When production is large, the effect is destabilizing, unless production is symmetric. The Authors suggest that then the asymmetry, rather than the strength, is destabilizing. This is interesting in light of Bascompte et al. (Science 2006), who argued precisely the opposite. This part could be expanded, and the analysis contrasted with that of Bascompte et al. (who also used LV-like dynamics).*

We concluded that both mutualistic interaction asymmetry *and* strength can be important in determining stability in consumer-producer-abiotic-resource systems. Drawing from the model we analyzed in Section 1.2.1, our evidence for this was two-fold. First, we found there is a threshold on interaction strengths, such that below this threshold stability is guaranteed. This threshold isn’t guaranteed to be necessary, but we provided evidence (shown in Figure 3) that it isn’t superfluous, as there are certainly cases where asymmetric, strong interactions lead to unstable feasible equilibria. On the other hand, in the limit of completely reciprocated (i.e. symmetric) interactions, we were able to prove that the positive equilibrium will be stable. So while we don’t yet know how to combine these two insights analytically with full generality, we know that both strength and symmetry are important.

This indeed is at odds with our understanding of local stability in other models of mutualism, and so we thank the reviewer for flagging up this issue and helping us to convey the interpretation even more clearly. The analysis of Bascompte et al. (Science 2006) is based on Lotka-Volterra equations—the kind of model we are arguing may not be so appropriate for microbial communities (though of course these equations may be a highly appropriate starting point for modeling the kind of plant-pollinator mutualism considered there).

The central result in Bascompte et al states that if the product $\alpha\beta$ of the average effect of pollinators on plants (α) and the average effect of plants on pollinators (β) satisfies a bound, the resulting system will be stable:

$$\alpha\beta < \frac{\text{intraspecific competition}}{\text{system size}}. \quad (1)$$

The critical threshold is related to the strength of intraspecific competition and the system size. And so even though the interactions in Bascompte et al are highly structured, the result is extremely reminiscent of the kinds of bounds that the Reviewer and others have identified for random Lotka-Volterra systems. Basically, if the overall interaction strengths are too large relative to the (stabilizing) effect of intraspecific competition, then a positive equilibrium will be unstable.

So how does this relate to asymmetry? Bascompte et al quite rightly conclude that if some interactions are very strong (e.g. the effect of plants on pollinators, β is large) then stability will require that other interactions (α) are weaker, to compensate. So we would say that it is not the asymmetry itself that confers stability in this model, but really a bound on overall interaction strength, meaning that if some interactions are large, then (in keeping with other Lotka-Volterra analyses) others will have to be weak.

Our result actually seems quite different—mutualistic interactions can all be very strong, but reciprocity will still guarantee stability. Ultimately, this comes down to the quite different assumptions in our model. So in a way, the differences with Bascompte et al support our claim that viewing mutualism through the lens of abiotic resources is worthwhile precisely because the results differ from LV-type analyses.

We thank the reviewer again for this opportunity to make the point more clearly, and we have both cited the suggested paper, as well as including the following text in Section 1.2.1 to reflect this discussion:

“A second example is when the production matrix is purely symmetric. In this case, feasibility alone is sufficient to guarantee local stability of the equilibrium solution, with no further restrictions on production. We call this case reciprocal, because symmetry of P ensures

that each (specialized) consumer gives exactly as much of each other consumer’s preferred resource as they receive from that consumer. Fig.3 demonstrates this for some particular examples. The importance of reciprocity for stability seemingly contrasts with analysis of mutualistic interactions using direct interactions between species—for example in the case of plant-pollinator networks [Bascompte et al]. In these highly structured communities, it was found that strong effects of plants on pollinators (or vice versa) must come with weak interactions of pollinators on plants, or else the resulting community equilibrium would tend to be unstable. Even though these communities differ from the random interactions considered by [Allesina et al] and others, the same principle lies behind these results: if overall interaction strengths exceed a bound determined by intraspecific regulation, then a positive equilibrium can be unstable. In [Bascompte et al], this bound means that stability requires a balance of strong and weak interactions, leading to asymmetry, rather than reciprocity. In our model, our finding that the asymmetry in mutualistic interactions, is destabilizing rather than stabilizing demonstrates that there are critical insights to be gained from the analysis of this consumer-producer-resource community.”

3. *The distribution of eigenvalues in Figure 1 is quite interesting. My impression is that the real eigenvalues spreading on the x-axis must be strongly correlated with those of the diagonal matrix $-CS^*$. The "wings" of the dragonfly seem to be similar to those found for purely bipartite systems by Grilli et al. (Nature Comm, 2016), shifted to the left. In particular, the method of Grilli et al. could be used to approximate the span of the "wings".*

We thank the reviewer for this citation. We’ve now included a citation to the Grilli et al paper in Figure 1. We agree that a comparison between our ‘dragonfly’ distribution of eigenvalues and those earlier results may be interesting (even though it is fair to say the latter is based on a different set of differential equations). We also added some comments in the text on the interpretation of the full spectrum, though we think a full analysis of its shape likely lies beyond the scope of the current paper.

4. *The form taken by the Jacobian is very similar to the case studied by Case and Casten (American Naturalist, 1979). I believe the Authors should comment on the similarities/differences with this study. Also the study of Goh (also American Naturalist, 1979) should be discussed.*

Indeed—the Jacobian is very similar to the Case and Casten case, which we had cited in Section 1.1.1, and we agree with the Reviewer’s comment that this is deserving of more discussion. In essence, the Case and Casten result is for a ‘horizontal’ community comprising one group of organisms that are purely predators, and another group of reproducing organisms that is purely prey for the first group. The similarity of the Jacobian in this model of consumers and abiotic resources suggests that this horizontal structure of the community

is important for local stability. On the other hand, we argue in our introduction that the case of microbes and abiotic resources this horizontal structure may be the most likely type of community where such horizontal structure actually holds. To reflect this discussion, we modified the following text at the end of Section 1.1.1

“These results bear comparison with earlier calculations for the stability of systems of biotic consumers and biotic resources. In these systems, resources themselves can grow and compete with each other, and can be eaten by the consumers. When treated as a set of Lotka-Volterra equations, in those cases feasibility also implies stability. The similarity with our results suggests that the horizontal structure here (i.e. a clear division of the system into consumers and resources) is key element for the local stability guarantee to hold. In large systems of consumers and biotic resources it might be seen as artificial to completely separate consumers and resources into two groups. On the other hand, in microbial systems with a clear biological distinction between abiotic resources and biotic consumers, this separation is very natural, and so our result applies unambiguously and generally. ”

We have also cited both Case and Casten and Goh (1979) in the discussion section. Both papers focus on global stability for systems of Lotka-Volterra equations (the latter proving that locally-stable, mutualistic Lotka-Volterra system will also be globally stable). We see this as an important open question in these systems involving consumption and exchange of abiotic resources, and added text to this effect in the second-to-last paragraph of our discussion as a future direction:

“...extending the more general analyses of global stability that have been developed in the case of Lotka-Volterra competition and mutualism...”

*Overall, I greatly enjoyed reading the manuscript.
Stefano Allesina*

We again thank the reviewer for their nice comments and helpful suggestions for edits.

Response to Referee 2's Comments for the Authors

The subject of this paper is the stability of equilibrium solutions to systems of resource-consumer equations modelling microbial communities. Many classical results treat ecological as pairs of species that are directly interacting, either via competitive, mutualistic, predator-prey interactions, or a combination thereof. In reality, competitive and mutualistic interactions are frequently not direct but mediated via, respectively, consumption of a shared resource, or production by one species of a resource that is consumed by another. This paper takes the classical approach of random matrix models for ecological communities and investigates the effect of treating interactions not as direct pairwise interactions, but explicitly as consumer-resource interactions. The authors show that this changes some of the results from the classical theory. Specifically, they show that there is no loss of stability with increasing complexity when interactions are only via consumption. When there are also interactions via production, this can cause a loss of stability and this depends on factors such as the asymmetry of the effective mutualistic interactions induced by crossfeeding.

We thank the Reviewer for the clear summary.

Overall, I found the paper to be clearly written, the results to be and novel and sound and the interpretation appropriate. The authors strike a reasonable balance of making the paper accessible to a non-specialist yet including enough detail in the Supplementary Material. I recommend publication and have only minor comments:

We also thank the reviewer for these overarching comments.

- 1. The authors focus on the local stability of feasible equilibria, which they define as solutions where all species have strictly positive density. There are also a raft of solution in which one or more species has zero density. These can be accommodated within the theoretical framework as they can be viewed as a strictly positive solution of a subsystem of the original model, with the zero species removed. However, this masks an important question which is how does species coexistence depend on model parameters (complexity). If an increase in complexity tends to cause species extirpations (via a transcritical bifurcation in which a species' equilibrium density becomes zero), this will not be detected by the current analysis. This is a common feature of this type of theoretical framework, but the authors should at least add a discussion of this limitation of their approach.*

Our analysis in the first draft of this manuscript focused on cases with N consumers, N resources, and we focused on questions related to local stability for feasible solutions, where

all consumer and resource densities were positive. If system complexity is defined in terms of how complex (in whatever sense) is the matrix of consumer preferences, C_{ij} , then the answer is that stability does not depend on this. C_{ij} could be diagonal, random, or structured in some arbitrary way. Following on from this, the Reviewer is then asking an excellent question—what happens if one or more of these consumers is locally extirpated. Is the remaining assemblage still guaranteed to be stable?

To address this question, we considered first a more general case of N_S consumers, and N_R resources, now referenced in Section 1.1 of the main text, with full derivations of results in Section 3.5 of the methods. Before sending any consumers to zero density, we first note (in section 3.5.2) that feasible solutions still imply stability, so long as the number of resources is greater than or equal to the number of consumers.

We then consider the Reviewer’s suggestion. If e.g. all but one consumer are at positive densities, and one is at zero, what are the conditions for stability? First, we identify that there are conditions (involving the matrix C and the remaining, non-zero consumers) that are necessary for this system of equations to have finite, non-zero solutions for all resource densities. Second, we show that the resulting equilibrium of $N_S - 1$ positive consumer abundances, and one zero consumer abundance, can still be stable, though it is no longer guaranteed. The condition for stability is now simply that the extirpated species has a negative growth rate when rare, in this background—i.e. that the new equilibrium is uninvasible by the extirpated species. In section 3.5.3, we state mathematically (in terms of the non-zero resource and consumer densities) what this condition is, and we note that there is a natural generalization for multiple extirpated species.

2. *The text on p2 should be labelled as an Introduction section.*

We made this change.

3. *In eq. 2 the notation C^{-T} is ambiguous. I assume it means the inverse of the transpose of C . If so this can be unambiguously and better denoted as $(C^T)^{-1}$*

Fixed.

4. *The definition at top of p4 of what is meant by the diag notation should be moved to immediately after eq. 2 where the notation first appears.*

We moved the text to appear earlier in order to clarify this notation immediately after Eq (2).

5. *P4: If all eigenvalues of this Jacobian are negative ? If all eigenvalues of this Jacobian have negative real part*

Thanks for pointing out this typo—now fixed.

6. *p4 Independent of the form of the matrix C. I am guessing it is assumed that all elements of C are non-negative. This assumption should be explicitly stated either here or at eq. 1*

That is correct. We assume that the entries of C are all non-negative, as each of entry is interpreted as a (per capita) rate. To make this assumption clearer we edited the text following Eq (1) to read:

“while the non-negative quantity C_{ij} is the rate of consumption of resource i by consumer j , per unit consumer and resource.”

7. *I found the 2nd paragraph of p4 confusing. How are the earlier results less valid than the current ones? Some more detail in the argument is needed here.*

We have reworded this paragraph to avoid confusion. Our point was not that these earlier results for particular Lotka-Volterra systems are not valid—if the community structure is as assumed in those papers then feasibility implies stability. On the other hand, assuming that a real community of biotic species can be split into a subset of pure predators and a subset of purely prey species might be seen as somewhat artificial. And we know that for LV systems more generally, feasibility does not imply stability. To clarify, we reworded this text:

“These results bear comparison with earlier calculations for the stability of systems of biotic consumers and biotic resources. In these systems, resources themselves can grow and compete with each other, and can be eaten by the consumers. When treated as a set of Lotka-Volterra equations, in those cases feasibility also implies stability. The similarity with our results suggests that the horizontal structure here (i.e. a clear division of the system into consumers and resources) is key element for the local stability guarantee to hold. In large systems of consumers and biotic resources it might be seen as artificial to completely separate consumers and resources into two groups. On the other hand, in microbial systems with a clear biological distinction between abiotic resources and biotic consumers, this separation is very natural, and so our result applies unambiguously and generally. ”

8. *In the text discussing Fig. 1 it would be helpful to provide a description of the biological interpretation of the 4 different types of structure tested for the matrix C.*

To clarify this, we added some new text to the Fig 1 caption:

“Both left and right hand panels therefore show a gradient from generalism to specialism, but the right-hand case assumes that there is an unambiguous spectrum of similarity for resources, and that species that can consume a given resource also tend to consume similar resources.”

In summary, we showed both cases because we think that it is reasonable that for some kinds of resources this idea of a spectrum of similar resources is reasonable (and is common in the consumer-resource literature, where often a set of resources is imagined to line up along an axis). This could be the case if multiple ‘similar’ resources require overlap in metabolism. On the other hand, there could also be biologically-reasonable circumstances where each of a set of resources requires a completely distinct set of pathways. This led to the idea of treating the off-diagonal elements of C_{ij} as random.

9. *Following eq. 5 state that element of P are non-negative*

We clarified this at the beginning of Section 1.2 by adding the text:

“and a non-negative matrix, P , representing the production of resources by the consumers:”

10. *The sufficient (but not necessary) condition in eq. 8 comes from an upper bound on the real part of the eigenvalues. Upper bounds can be misleading! For example, taken at face value eq. 8 might be taken as suggesting that increases in P or decreases in the RHS of eq. 8 will tend to destabilise the system. But it perfectly possible for the upper bound on the eigenvalue to be increasing as P increases, but for the eigenvalue itself to be decreasing. On the whole, the authors use appropriately cautious language in interpreting their result in eq. 8. But it would be worth adding an explicit caveat that it is quite possible for systems to become more stable (i.e. the dominant eigenvalue’s real part to decrease) not less as production rates increase.*

We think this is a fair comment, and we tried to be cautious about the relevance of this upper bound. Indeed, the two examples we focus the most on (zero production of resources and reciprocal production of resources) violate this bound, and yet still in these two cases feasibility implies stability. Indeed, it could have been the case that our sufficient bound was entirely superfluous, and feasibility always implies stability. However, our Fig 3 shows that this is not the case. Combining specialist consumers with random production matrices P_{ij} that violate our bound generically leads to unstable equilibria. The bottom line is that we know our bound is sufficient, and not necessary, but we also don’t think that it is irrelevant.

We revised the text following this bound, and we hope that the overall impression given is now clearer:

“These results are reminiscent of constraints on stability for randomly-drawn, pairwise interactions between species. In that case, mutualistic interactions tend to be more destabilizing than competitive interactions. Are we just recapitulating results that are qualitatively already understood via pairwise interactions? We emphasize that here our bound is sufficient, but not necessary for feasibility to imply stability, and there are cases where the bound above

is not necessary for stability. One example is when production vanishes altogether, and we are back to our earlier, more general result that feasibility always implies stability.

A second example is when the production matrix is purely symmetric. In this case, feasibility alone is sufficient to guarantee local stability of the equilibrium solution, with no further restrictions on production. We call this case reciprocal, because symmetry of P ensures that each (specialized) consumer gives exactly as much of each other consumer's preferred resource as they receive from that consumer. Fig.3 demonstrates this for some particular examples. The importance of reciprocity for stability seemingly contrasts with analysis of mutualistic interactions using direct interactions between species—for example in the case of plant-pollinator networks. In these highly structured communities, it was found that strong effects of plants on pollinators (or vice versa) must come with weak interactions of pollinators on plants, or else the resulting community equilibrium would tend to be unstable. Even though these communities differ from the random interactions considered by and others, the same principle lies behind these results: if overall interaction strengths exceed a bound determined by intraspecific regulation, then a positive equilibrium can be unstable. In [Bascompte et al], this bound means that stability requires a balance of strong and weak interactions, leading to asymmetry, rather than reciprocity. Our finding that the asymmetry in mutualistic interactions can destabilizing rather than stabilizing demonstrates the quite different insights to be gained from the analysis of consumer-producer-resource communities. ”

Response to Referee 3's Comments for the Authors

Understanding the conditions in which microbial communities will be stable is an important problem in community ecology. In this paper, the authors present analytical results of a resource-consumer model and its variant. In its basic form, the model is defined in Eqs. (1) and accounts for different consumer preference (C_{ij}), different resource influx rates (ρ_j) and consumer death rate (μ_i). In the basic model, the species do not produce resources. In its variant (Eq. 5) mutualistic interactions are considered. The authors show that in the basic model 1): Feasible solutions are stable regardless of the dimension of the system N (feasible solutions mean that there is a fixed point with positive values for the number of resources and consumers). 2) The volume (= "number", aka structural stability) of feasible solutions may (= for particular parameter instances) decrease even when species similarity is decreased (Fig. 2). Finally, the authors incorporate cross-feeding and derive sufficient conditions on the matrix of cross-feeding interactions for which the feasible solutions will be stable (Eq 8).

The significance of these results is the fact that they differ from those obtained in pairwise-interaction models (see e.g. Allesina et al. [6]) where the resource-consuming interactions are not accounted for. I am not an expert on this class of theoretical models, so I am not able to judge the novelty of some parts of the work (although the first part of the paper feels like something that should have been done before it would be good for the authors to be more clear about this). I thought that the model was a generally reasonable model and was analyzed in a rigorous way, although I often felt that intuition was not provided for the reader. In addition, the figures were not very helpful in supporting the authors paper. Given the issues specified above I sometimes had difficulty fully appreciating the paper, but at the same time I believe that this sort of resource-explicit model has the potential to provide tremendous insight into microbial communities.

We thank the Reviewer for the excellent summary of our paper, and we hope that addressing the points raised below (and by the other reviewers) will help to convey the intuition for and significance of the results.

MAJOR POINTS

1. - *It is my understanding that the authors' results are valid only in the case N resources = N consumers. Is that true? If so, I'd prefer to have this point expressed more clearly in the introduction, and perhaps show in the Discussion section whether or not the results generalize to the case $no. resources \neq no. consumers$. This is particularly important given the history*

of the field in which these issues have received considerable attention.

Our analysis in the first draft of this manuscript did indeed focus on cases with N consumers and N resources. We have now stated this clearly in Section 1.1.1, both first and second paragraphs.

To then address the natural question raised by the reviewer of what happens in more general cases, we considered a model with N_S consumers and N_R resources, also referenced in Section 1.1.1 of the main text, with full derivations of results in Section 3.5 of the methods. Our central result in this analysis (contained in section 3.5.2) is that feasible solutions can still imply stability. However, this is only true so long as the number of resources is greater than or equal to the number of consumers. In these cases, feasibility implies stability, but for fewer resources than consumers this result does not hold. As the Reviewer is perhaps hinting, this outcome is probably to be expected given the expectation that coexistence will (in many cases) require at least as many distinct resources as competing species.

2. - *I feel that the 'Results' section does not add much, beyond stating the results. For example, it is not very helpful to show Eq. (3) or (4) because those formulae do not become clear until one gets to the 'Method' section. In contrast, the Method section is highly technical and, in its current stage, not suitable for the average reader. I'd recommend the authors to expand the 'Results' section and provide some intuition about their results.*

We agree with the reviewer that there is a tricky balance here in terms of readability and information. We do not think that the full content (or even the majority) of the Methods section would be appropriate in the main body of the Results section—it is there primarily so that readers can pick out specific results and work through our approach (or extend it, etc). These details/derivations would probably obscure the clarity of the results section, where we tried to state the various outcomes succinctly.

However, we take the point that biological interpretations have to be made as clear as possible, and in some places being less succinct may be helpful. Combining some of the Reviewers other comments, and our responses to other reviewers, we now have several pieces of additional text in the Results section reflecting biological interpretations:

- The importance of the relative numbers of resources and consumers, what happens to our stability statements when these numbers are different, and how this relates to classical ecological intuition (Section 1.1.1)
- What happens if a consumer is knocked down to zero abundance, in which case we find the remaining consumers coexist stably so long as this new equilibrium is uninvasible by the extirpated species (1.1.1)

- We give a clearer explanation of the limitations of modeling substitutable resources, and the functional dependence of growth rates for more general forms of (co-)limitation (1.1)
- We interpret (in broad outline) the implications of the overall shape of the eigenvalue spectrum around an equilibrium involving only consumers (1.1.1)
- We clarified the importance of the functional form of structural stability on the matrix C (1.1.2)
- We give a fuller explanation of the interpretation of our results for mutualistic interactions, alongside a clearer comparison with results obtained using the Lotka-Volterra formulation of mutualistic interactions (1.2).

In summary, we think that our three central results are now presented with clearer biological intuition in the Results section. I.e. that (a) consumers alone will coexist at a locally stable equilibrium, so long as there are at least as many abiotic resources as consumers (b) structural stability depends in a non-trivial way on consumption preferences, and specifically not in what would be the obvious/naive way (something like the average of overlaps in consumer preferences) and (c) that in the case of production, if species reciprocate with each other cooperatively, positive equilibria will still be locally stable. We hope the Reviewer agrees that these are now conveyed with greater clarity.

3. *-The authors model assumes that resources are made in proportion to the species abundance rather than their growth (or what they are growing on). This is a major simplifying assumption, so the authors should perhaps be more clear about it.*

This is a reasonable comment, and we agree with the Reviewer that there are certainly more general forms of production that would also be reasonable. We think our interpretation of the production terms is also reasonable—either as a species-specific form of leakage, or as a form of recycling where some portion of consumer biomass is returned to the common pool during mortality. But likely not the most general case.

Reviewer 1 had a similar comment, and to address both suggestions, we modified text in two places. Below the introduction of the equations involving leakage (in the subsection titled ‘Mutualistic Interactions’) we added to and modified the following paragraph:

“Our approach again makes use of mass action principles, and we note that we consider production of resources to remove biomass from the consumer density, meaning that terms corresponding to this leakage appear in both equations. We have also chosen a form for production that depends solely on the density of species that are present in the system, and not on what resources those species are using to grow. This may be a reasonable approximation

for some processes, for example the production of various intermediates of the TCA cycle, the production of which are not substrate-dependent. But in other cases, we may need to allow for a more general form for production that depends both on the species that are present, *and* the specific resources they are consuming. Another interpretation of the same terms would be as a kind of recycling—where mortality doesn’t just remove consumers from the system, but also returns some portion of their biomass to the common resource pool.”

We also thought it was worthwhile outlining these choices in the introduction, and so we added this sentence to the fourth paragraph of the introduction:

“We introduce a model for the exchange of resources, where microbes can both consume and now also produce resources, and choose a form for these equations that is interpretable as either a species-specific leakage of resources, or as a kind of recycling of biomass following mortality.”

4. *I do not think that Fig. 2 is very helpful in supporting the authors’ argument. Maybe it’d be better to compute (numerically) and the volume V_{mu} as a function of the number of species (for the parameter instance for which the effect is observed)? It is not immediately clear from the figure that the blue volume is smaller than the red one.*

This is a fair comment, though the mathematical results in the Methods section back-up the visual impression that the blue volume is smaller. Wrestling with how to visualize structural similarity for 3+ species is definitely non-trivial, and we can’t really get any simpler—our example doesn’t work with fewer than 3 consumers.

We are mindful though that even if readers agree that this exemplifies the idea that structural similarity *can* depend in a non-trivial way on consumer similarity (in terms of consumption preferences), the intuition as to why this is, or how it generalizes, might not come across as clearly as it should have done. To clarify the interpretation here, we have modified the text to emphasize two issues. First, the form of V_{μ} mathematically does not depend solely on the pairwise similarities of each pair of consumers. That kind of dependence might have been the biological expectation, but it is not borne out by our analysis.

Second, (and in part this is our reasoning for showing Fig. 2) there are clear cases where the structural similarity (counterintuitively) decreases with the pairwise similarity of a focal pair of species. To emphasize that generically there is no simple dependence on pairwise similarity, we added text (detailed below). Putting these two ideas together (the general result and the specific examples) we changed the last two paragraphs of Section 1.1.3 to read:

“This contrasts with what perhaps might be the intuitive determinant of structural stability—something like the average of pairwise species similarity (say, defined in terms of the overlaps

of two consumers’ preferences). In contrast, the determinant here does not depend in a simple way on the similarity of any two species—structural stability depends on C as a whole, rather than being a function just of pairwise similarities.

“We can still ask how this volume changes as we make consumers more or less similar in terms of the resources they use, represented mathematically by the inner product of the columns of C . For example, in the special case where all species begin equally similar, then a uniform decrease in their similarity leads to a larger volume for the parameter space, and greater structural stability (detailed in Methods 3.3.1). This is in agreement with the ecological intuition that it is ‘easier’ for more dissimilar species to coexist. However, this is not the general case. Independent of the size of the system, there are some contexts where a decrease in the similarity of any pair of species will lead to less, not greater, structural stability. Fig. 2 shows a specific example, for three species, where a decrease in the similarity of one pair of species (while keeping other pairwise similarities fixed) leads to a decrease in structural similarity. What is the general lesson here? The biological interpretation is that changes in structural stability as we change the similarity of any two species depend (in general) on all of the other consumption preferences. This reiterates our point above, that the determinant in V_μ does not depend in a trivial way on pairwise comparisons, and consequently the structural stability of systems of consumers and abiotic resources does not depend in a simple way on pairwise species similarities expressed in terms of consumption preferences.”

5. *In general, it seems that the figure purpose is to exemplify the claims by instantiating the model for certain parameter values. This is OK, but not sufficient for me to believe that the claims are correct. Reproducing the calculations might not be possible for every reader given that they are highly-technical. I think the authors should concentrate their efforts in either finding a biological explanation, or to corroborate their results further with numerical simulations.*

We agree with this point made by the Reviewer—the figures clearly represent only a small subset of all possible examples. We note that in Figures 1 and 3 (as stated) we do aggregate results across multiple combinations of consumer and resource densities, but it is true that in each case we only consider a handful of C and/or P matrices.

Our intention was not that these figures would prove our mathematical results, and the Reviewer should not rely on them as a proof. Indeed, we would have to admit that adding further numerical simulations would not achieve this either, no matter how many cases we looked at. Instead our intention is that the figures will achieve two things: (a) provide a check for reviewers interested in exactly how our proofs manifest themselves in a specific example and (b) as an illustration that there are more general properties of these spectra

that are worthy of further investigation (discussed elsewhere in our response to the Reviewer's comments).

So while in some ways one could argue that we should just state the mathematical results we've proved, and that would be enough, we do think that the figures will help with the reader's intuition in interpreting these results, and may suggest interesting future work. For these reasons we have retained the figures as is, though we have taken advantage of the Reviewer's comments elsewhere on how best to clarify them. We hope also that some of our additional text on interpreting results (discussed above) will aid with the biological interpretation.

MINOR POINTS

- (a) - *Sec. 1.1 'characterizing' typos*

Thanks—we fixed this typo.

- (b) - *I would recommend to add ' $i = 1, \dots, N$ ' next to Eq. (1).*

Done.

- (c) - *Sec. 1.1 par 2: The authors may want to explain 'Liebig's law of minimum' and the other rules that define limiting factors of microbial growth. I think this sentence contains too much jargon.*

We agree, and rephrased this paragraph as follows, to give a more explicit and intuitive picture for how these forms of colimitation would change Eq 1:

“Finally, we note that this is a model for substitutable resources, and while there may be families of resources which *are* to some extent substitutable (for example different carbon sources) the general picture for microbial consumers is likely colimitation by multiple, qualitatively different types of resource. In many cases, we might expect that only one of these colimiting resources is rate-limiting (roughly, the rarest in a given environmental context), and this assumption leads to Liebig's law of the minimum, where growth rate of a consumer depends only on this single resource. In other contexts, two or more resources may be limiting in any one context, in which case growth rate has typically has been modeled as proportional to the product of these limiting resources, and termed multiplicative colimitation. Beyond these phenomenological approximations of colimitation, there might be still more general functional dependencies. Taking these possibilities into account, our analysis of substitutable resource consumption and production here may be seen as a starting point for these more general cases,

and a good approximation for circumstances where a single ‘type’ or family of resources is rate-limiting. ”

- (d) *Fig. 1 and Fig. 3 It’d be helpful to label the axes of the insets. For example, in Fig. 1, it is my understanding that the insets represent $N \times N$ matrices whose i, j - th element defines preference of i -th consumer for j -th resource (or the other way around)? Blue indicates greater preference than white (or the other way around)? Are they just the C matrices?*

We tried labeling these inset axes, though the result was (to our eyes) visually a bit confusing. However, we take the point that the meaning of these grids was not sufficiently clear. Hence, we have included in the figure captions a clearer explanation of these insets. In the case of Figure 1, we added:

“The form of the consumer preferences is shown inset in blue, where each row represents a distinct resource, each column represents a distinct consumer, and darker blue indicates a higher rate of consumption”

While in Figure 3, we added:

“Plots show the density (colored from red to blue) of eigenvalues for the Jacobian matrix at this equilibrium, defined using a fixed, diagonal matrix of consumer preferences. This is shown inset in blue, where each row represents a distinct resource and each column represents a distinct consumer; blue squares indicates a non-zero rate of consumption. We also consider a fixed, more general matrix of production rates (inset in green). Again, each row represents a distinct resource, each column represents a distinct consumer, and darker green indicates a higher rate of production.”

- (e) - *The main result conveyed in Fig. 1 is the fact that $Re[\lambda] < 0$. I was wondering if it’d help to plot $\max Re[\lambda]$ against a control parameter that spans from generalism to specialism.*

We thank the reviewer for this suggestion. However, it isn’t actually clear exactly what effect the gradient from generalism to specialism (at least in the regimes we consider in the figure) has on how close to instability the equilibrium is (i.e. how large is the largest real part of any of the eigenvalues). In fact, we have not identified precisely what combination of parameters does control this. But we completely agree with the reviewer that this is an important question, and clearly worthy of further investigation. Hence, have added the following text to the discussion section, paragraph 3:

“Exact or approximate solutions for the full spectrum, including what controls its overall shape and the size of its largest eigenvalues, will shed light on the dynamics near equilibrium.”

(f) - *Fig. 1 What are the implications in the dynamics in having a 'dragonfly' shape of the spectrum compared to the classical ellipses?*

Again, while we emphasize that understanding stability itself (as a binary question) is already biologically relevant, we completely agree that the form of these eigenvalue spectra is of interest. In part, this is why we felt it important to illustrate our theorems/results with specific examples—these examples demonstrate visually cases where equilibria are stable vs unstable, but they also might point the way to more general, future analyses of the eigenvalue spectrum.

In terms of what we can say at this point, we point out two important features of these spectra. Clearly the wings indicate the potential for oscillations near an equilibrium, though these complex eigenvalues are also there (in some form) in the classical circular/elliptical shape. So in a way what may be more interesting is the relative large number of real eigenvalues, and a relatively high density near zero (which will lead potentially to long relaxation times). We do not think that a full analysis of this distribution is within the scope of our current paper—but we take the Reviewer's point that this is worthy of some discussion. Hence, we added text to section 1.1.1, paragraph 1:

“ These plots demonstrate visually the form of the spectrum across a range of cases (both generalist and specialist consumers, and ordered and unordered resources). Beyond the fact that the largest eigenvalue always has negative real part, as our results state, we also note that the spectra have a characteristic ‘dragonfly’ form. The wings of the dragonfly will lead to the potential for oscillatory behavior around this equilibrium, while the typically large density of eigenvalues *near* zero imply that some types of perturbation away from these equilibria will have long relaxation times.”

(g) - *Fig. 1 and 3: could the authors add a legend for the green-red colors used?*

Thanks for this suggestion—we have added a legend to each panel in Figures 1 and 3.

REVIEWERS' COMMENTS:

Reviewer #1 (Remarks to the Author):

I found the responses to my comments clear and complete. I appreciate the changes made to the manuscript.

I do not have further comments.

Reviewer #2 (Remarks to the Author):

The authors have done a very nice job of addressing Reviewer comments on the previous version and I now recommend publication.

Reviewer #3 (Remarks to the Author):

I believe that the authors have adequately addressed the questions of the referees, and that the manuscript is now ready for publication.